# Tight Stability, Convergence, and Robustness Bounds for Predictive Coding Networks

## Abstract

Energy-based learning algorithms, such as predictive coding (PC), have garnered significant attention in the machine learning community due to their theoretical properties, such as local operations and biologically plausible mechanisms for error correction. In this work, we rigorously analyze the stability, robustness, and convergence of PC through the lens of dynamical systems theory. We show that, first, PC is Lyapunov stable under mild assumptions on its loss and residual energy functions, which implies intrinsic robustness to small random perturbations due to its well-defined energy-minimizing dynamics. Second, we formally establish that the PC updates approximate quasi-Newton methods by incorporating higher-order curvature information, which makes them more stable and able to converge with fewer iterations compared to models trained via backpropagation (BP). Furthermore, using this dynamical framework, we provide new theoretical bounds on the similarity between PC and other algorithms, i.e., BP and target propagation (TP), by precisely characterizing the role of higher-order derivatives. These bounds, derived through detailed analysis of the Hessian structures, show that PC is significantly closer to quasi-Newton updates than TP, providing a deeper understanding of the stability and efficiency of PC compared to conventional learning methods.

## 1 Introduction

The successes of artificial intelligence (AI) over the past decade have been primarily driven by deep learning, which has become a cornerstone in modern machine learning. Despite these remarkable achievements, there has been a resurgence of interest in exploring alternative paradigms for training neural networks Zador et al. (2022); Hinton (2022). This renewed interest stems from the inherent limitations of standard backpropagation-based training (BP) Rumelhart et al. (1986), such as high computational overhead, energy inefficiency, and biological implausibility Lillicrap et al. (2020); Ororbia et al. (2024). As a result, there has been a revival of energy-based learning methods that were initially popularized in the 1980s, such as Hopfield networks Ramsauer et al. (2020), equilibrium propagation Scellier & Bengio (2017), contrastive Hebbian learning Høier et al. (2023), and predictive coding Salvatori et al. (2023). These models, which minimize a general energy functional Ororbia et al. (2024), employ local update rules that are well-suited for both spiking neural networks Ororbia (2023); Lee et al. (2024) and neuromorphic hardware implementations Kendall et al. (2020).

Among these approaches, predictive coding (PC) stands out due to its intriguing theoretical properties and practical performance. Originally developed to model hierarchical information processing in the neocortex Rao & Ballard (1999); Friston & Kiebel (2009), PC has shown remarkable performance in a wide range of machine learning tasks, such as image classification and generation Ororbia & Kifer (2022); Pinchetti et al. (2024), continual learning Ororbia et al. (2020), associative memory formation Salvatori et al. (2021); Tang et al. (2022), and reinforcement learning Rao et al. (2023). In addition to its practical applications, recent work has focused on analyzing PC from a theoretical standpoint, examining its stability and robustness properties Alonso et al. (2022); Song et al. (2023). Prior investigations have drawn parallels between PC and other learning frameworks, such as BP and target propagation, but have not provided rigorous and tight bounds on the stability and robustness of PC's updates Song et al. (2020); Salvatori et al. (2022). Existing bounds are often based on first-order approximations Millidge et al. (2022), overlooking the impact of higher-order derivatives, which are crucial for understanding stability in non-linear systems.

A rigorous theoretical study of stability is essential for ensuring convergence to fixed points and robustness under various perturbations, which is crucial for applications such as noisy inputs or uncertain synaptic parameters. Stability has been shown to enhance the performance of neural networks in complex scenarios, e.g., sequence length generalization Stogin et al. (2024), symbolic rule extraction Dave et al. (2024); Mali et al. (2023), and stable prediction dynamics Chang et al. (2019); Dai et al. (2021); Haber & Ruthotto (2017). In biologically-motivated learning, where models are deployed on hardware platforms susceptible to noise, understanding stability is even more critical Kendall et al. (2020); Scellier et al. (2024). Hence, in this work, we leverage dynamical systems theory to rigorously analyze the stability and convergence properties of predictive coding networks (PCNs). Our contributions are as follows:

- **Derivation of Stability Bounds:** We derive tight stability bounds for PCNs using Lipschitz conditions on the activation functions. By leveraging these bounds, we establish that PCNs are Lyapunov stable, ensuring that their synaptic weight updates converge to a fixed point. This stability guarantees robustness to perturbations in the input, synaptic weights, and initial neuronal states, providing a formal foundation for the robustness properties of PCNs.
- **Quasi-Newton Approximation:** We demonstrate that PC updates approximate quasi-Newton updates for synaptic parameters by incorporating higher-order curvature information. This property implies that PCNs require fewer update steps to achieve convergence compared to standard gradient descent, owing to the incorporation of second-order information, which adjusts both the direction and magnitude of gradient steps.
- **Theoretical Comparison with BP and Target Propagation:** Using our developed framework, we derive novel bounds that quantify similarities between PC, backpropagation (BP), and target propagation (TP). These bounds explicitly capture the influence of higher-order derivatives, showing that PC is closer to quasi-Newton updates than TP, thereby providing a more refined understanding of its stability and efficiency relative to other learning schemes.

Our results establish a new theoretical foundation for PC as a robust and efficient learning framework, offering insights into its convergence properties and stability advantages over traditional learning methods. To this end, in the following section we will both introduce some background on PCNs, as well as defining concepts from the dynamical systems theory that will be used to prove our results.

## 2 BACKGROUND AND MOTIVATION

Predictive coding networks (PCNs) are hierarchical neural networks, structurally similar to multi-layer perceptrons, that perform approximate Bayesian inference and adapt parameters by minimizing a variational free energy functional Friston (2008); Salvatori et al. (2023); Rao & Ballard (1999). A PCN consists of $L$ layers with neural activities $\{x_0, x_1, \ldots, x_L\}$, where $x_0$ and $x_L$ correspond to the input and output signal, respectively. The goal of each layer is to predict the neural activities of the next layer via the parametric function $f(W_l x_{l-1})$, where $W_l$ is a weight matrix that represents a linear map, and $f$ is a non-linearity (or nonlinear activation function). The difference between the predicted and actual neural activities is referred to as the *prediction error*, formally defined and denoted as $\epsilon_l = x_l - f(W_l x_{l-1})$. During model training, the goal of a PCN is to minimize the total summation of (squared) prediction errors, expressed via the following variational free energy:

$$F = \sum_{l=0}^{L} E_l = \sum_{l=0}^{L} |\epsilon_l|^2, \tag{1}$$

where $E_l = |\epsilon_l|^2$ represents the local energy at neuronal layer $l$. This decomposition in terms of layer-wise energy functionals is what allows every update to be performed using local information only (i.e., only pre-synaptic and post-synaptic neural activity values), regardless of the fact that the updates rely on gradients. In supervised learning, the input layer $x_0$ is clamped to the data $d$ whereas the output layer $x_L$ is clamped to the target $T$. The free energy then decomposes into the loss at the output layer and the residual energy of hidden layers:

$$F = L + \tilde{E}, \quad with \quad L = E_L = |\epsilon_L|^2 \quad and \quad \tilde{E} = \sum_{l=1}^{L-1} E_l. \tag{2}$$

Minimizing the free energy functional $F$ in PCNs is then accomplished through two distinct phases: inference and learning. During the inference phase, the activations of the hidden neuronal layers,

denoted by $x_1, \ldots, x_{L-1}$, are iteratively updated so as to minimize $F$, while the activation values at the input layer $x_0$ and output layer $x_L$ are held fixed to the values of the input data and target output, respectively. The dynamics governing the inference updates for each hidden layer are as follows:

$$\Delta x_l = -\frac{\partial F}{\partial x_l} = -\epsilon_l + W_{l+1}^\top \left(\epsilon_{l+1} \odot f'(W_{l+1} x_l)\right), \quad \text{for } l = 1, \ldots, L-1, \tag{3}$$

where $\odot$ signifies elementwise multiplication. Following the convergence of the inference phase, the learning phase then commences. During this phase, the network's synaptic parameters, specifically the weights $W_l$ at each layer, are updated to further minimize the overall free energy as follows:

$$\Delta W_l = -\eta \left(\frac{\partial L}{\partial W_l} + \frac{\partial \tilde{E}}{\partial W_l}\right) = -\eta \left(\epsilon_l \odot f'(W_l x_{l-1})\right) x_{l-1}^\top, \tag{4}$$

where $\eta$ denotes the learning rate.

## 2.1 DYNAMICAL SYSTEMS BACKGROUND

In this section, we outline the key concepts of stability and robustness taken from dynamical systems theory; these are essential for analyzing the behavior of neural networks and their corresponding learning algorithms. Understanding these concepts further allows us to characterize the performance and reliability of different models in terms of their dynamical properties. Note that the definitions stated here are not rigorous, due to a lack of space needed to properly define all the details, and notation. Readers interested in the relevant formal definitions, which are notably used to be formally prove the theorems of this work, should refer to the supplementary material.

Firstly, we formally characterize what it means for a dynamical system to be stable.

**Definition 2.1.** *A dynamical system is said to be* stable *if, for any given $\epsilon > 0$, there exists a $\delta > 0$ such that if the initial state $x(0)$ is within $\delta$ of an equilibrium point $x^*$ (i.e., $|x(0) - x^*| < \delta$), then the state $x(t)$ remains within $\epsilon$ of $x^*$ for all $t \geq 0$ (i.e., $|x(t) - x^*| < \epsilon$ for all $t \geq 0$).*

We further consider comparing stability between two dynamical systems, defining conditions under which one system is considered more stable than another.

**Definition 2.2.** *Let $D$ and $D'$ be two dynamical systems with equilibrium points $x_D^*$ and $x_{D'}^*$, respectively. The system $D$ is said to be* more stable *than $D'$ if, for the same perturbation or initial deviation from equilibrium, $D$ either:*

*(i) Converges to its equilibrium point faster than $D'$,*
*(ii) Remains closer to its equilibrium point over time, or,*
*(iii) Has a larger region of attraction, indicating greater tolerance to perturbations.*

Next, we define the key dynamical systems analysis concept of a system reaching a fixed point.

**Definition 2.3.** *A point $x^* \in \mathbb{R}^n$ is called a* fixed point *or equilibrium point of the dynamical system $\dot{x}(t) = f(x(t))$ if $f(x^*) = 0$. The system is said to* converge to the fixed point $x^*$ *if, for initial states $x(0)$ in a neighborhood of $x^*$, the trajectories satisfy $\lim_{t \to \infty} x(t) = x^*$.*

Lastly, we address the rate of convergence between two stable dynamical systems, which can be used to measure convergence bounds for networks.

**Definition 2.4.** *Let $D$ and $D'$ be dynamical systems that converge to the same fixed point $x^*$. We say that $D$ converges* faster *than $D'$ if, for initial conditions close to $x^*$, the distance to the fixed point decreases at a higher rate for $D$ compared to $D'$.*

### 2.1.1 ROBUSTNESS TO PERTURBATIONS AND LYAPUNOV'S STABILITY

Beyond stability, *robustness* refers to a system's ability to maintain performance in the presence of disturbances or uncertainties. Robust systems can handle variations without significant degradation, which is critical for practical applications. This notion can be formally read as follows:

**Definition 2.5.** *A dynamical system is said to be* robust to random perturbations *if, for any small random perturbation $\eta(t)$ acting on the system, the perturbed state $x_\eta(t)$ remains close to the unperturbed state $x(t)$ with high probability for all $t \geq 0$. This implies that the system can maintain its desired performance despite small random disturbances.*

Similarly, robustness with respect to initial conditions ensures consistent system behavior even when starting from a range of possible initial states due to uncertainties or randomness.

**Definition 2.6.** *A dynamical system is said to be* robust to random initial conditions *if, given initial state $x(0)$ drawn from a probability distribution with bounded support, the system's state $x(t)$ remains close to the desired trajectory or equilibrium point $x^*(t)$ with high probability for all $t \geq 0$.*

Robustness is particularly important in artificial neural networks (ANNs) operating in real-world environments, where inputs and initial conditions can vary unpredictably. Applying these concepts to neural networks, Lyapunov's methods provide tools for assessing whether small changes in weights or inputs lead to small changes in outputs, which is crucial for ensuring reliable learning and inference. To this end, we now introduce the important definition of *Lyapunov stability*:

**Definition 2.7.** *An ANN is* Lyapunov stable *if, for any given $\epsilon > 0$, there exists a $\delta > 0$ such that if initial weights $W(0)$ are within $\delta$ of a trained equilibrium point $W^*$ (i.e., $|W(0) - W^*| < \delta$), then the weights $W(t)$ remain within $\epsilon$ of $W^*$ for all $t \geq 0$ (i.e., $|W(t) - W^*| < \epsilon$ for all $t \geq 0$). If, in addition, the weights converge to $W^*$ as $t \to \infty$, then the ANN is said to be* asymptotically stable.*

## 3 THEORETICAL RESULTS

In this section, we study the convergence and stability of PCNs by leveraging the framework of dynamical systems and Lyapunov stability theory established above. We establish a series of theorems that states that, if activation functions and their higher-order derivatives are Lipschitz continuous, PCNs converge to Lyapunov stable equilibrium points. These results are then used to compare convergence speed of PCNs against backpropagation (BP) updates. Note that such a condition on the activations excludes linear rectifiers (ReLUs) yet includes oft-used ones such as the logistic sigmoid, the hyperbolic tangent (tanh), and the newly proposed TeLU Fernandez & Mali (2024).

The following theorem provides a characterization of the convergence and stability of the PCN based on the properties of these types of functions (proof in the appendix).

**Theorem 3.1.** *Let $M$ be a PCN that minimizes a free energy $F = L + \tilde{E}$, where $L$ is the backprop loss and $\tilde{E}$ is the residual energy. Assume the activation function $f$ and its derivatives $f'$, $f''$, and $f'''$ are Lipschitz continuous with constants $K$, $K'$, $K''$, and $K'''$, respectively. Then, the convergence and stability of the PCN can be characterized by the bounds involving these higher-order derivatives.*

The above shows that if a PCN uses activations that are smooth and well-behaved (i.e., they and their first few derivatives change gradually without sudden spikes), then the network will converge reliably and maintain stability during training.

The next theorem crucially connects PCNs with continuous-time dynamical systems, facilitating analysis of their behavior with the plethora of tools of dynamical systems theory. Usefully, the result below effectively establishes that the dynamics of the neuronal layers (the E-steps Friston et al. (2008)) that make up PCNs do converge to equilibrium points

**Theorem 3.2.** *Let $M$ be a PCN that minimizes a free energy function $F = L + \tilde{E}$, where $L$ is the backprop loss and $\tilde{E}$ is the residual energy. Assume that $L(x)$ and $\tilde{E}(x)$ are positive definite, and their sum $F(x)$ has a strict minimum at $x = x^*$, where $F(x^*) = 0$. Further, assume the activation function $f$ and its derivatives $f'$, $f''$ are Lipschitz continuous with constants $K, K', K''$, respectively. Then, the PCN dynamics can be represented as a continuous-time dynamical system, and the Lyapunov function $V(x) = F(x)$ ensures convergence to the equilibrium $x = x^*$.*

This theorem 3.2 establishes the stability and convergence properties of a PCN that minimizes a **free energy** function, defined as $F = L + \tilde{E}$, with $L$ as the BP loss and $\tilde{E}$ as the residual energy. The conditions assumed are that both $L(x)$ and $\tilde{E}(x)$ are positive definite, meaning they are zero only at a unique equilibrium $x = x^*$ – where $F(x^*) = 0$ – and are strictly positive otherwise. The theorem further states that the free energy function $F(x)$ can be used as a Lyapunov function, $V(x) = F(x)$, to analyze the PCN's behavior as a continuous-time dynamical system. Since $V(x)$ is positive definite and decreases along the system's trajectories, it implies that the system will converge stably to the equilibrium point $x^*$ during training. Additionally, the smoothness assumptions on the activation $f$ and its derivatives (up to second order, i.e., $f'$ and $f''$) being Lipschitz continuous ensures that the updates are stable and do not lead to erratic behavior (proof/assumptions in the appendix). The

importance of this result is that it provides a formal guarantee for the convergence of PCNs, even in complex multi-layered networks. This makes it a powerful tool for understanding the theoretical underpinnings of PCNs and their use in various machine learning contexts.

The next result formally establishes that the parameter updates (M-steps Friston et al. (2008)), as governed by Equation 4, reliably converge to a fixed point of the variational free energy.

**Theorem 3.3.** *Let $M$ be a PCN that minimizes a free energy function $F = L + \tilde{E}$, where $L$ is the backpropagation loss and $\tilde{E}$ is the residual energy. Assume that $L(x)$ and $\tilde{E}(x)$ are positive definite functions and their sum $F(x)$ has a strict minimum at $x = x^*$. Further, assume activation function $f$ and its derivatives $f'$, $f''$, and $f'''$ are Lipschitz continuous with constants $K, K', K''$, and $K'''$, respectively. Then, the PCN updates are Lyapunov stable and converge to a fixed point $x = x^*$.*

This theorem provides a rigorous guarantee of stability and convergence of a PCN that minimizes a free energy functional $F(x)$. Note that this result assumes that both $L(x)$ and $\tilde{E}(x)$ are positive definite functions, meaning that they are zero only at the equilibrium point $x = x^*$ and strictly positive elsewhere. This ensures that their combination $F(x) = L(x) + \tilde{E}(x)$ also reaches its strict minimum value at $x = x^*$. Theorem 3.3 further requires that the activation $f$ and its higher-order derivatives $f'$, $f''$, and $f'''$ are Lipschitz continuous, which guarantees smooth changes in these functions without sudden spikes/oscillations. These Lipschitz continuity conditions, defined by the constants $K, K', K''$, and $K'''$, ensure that the PCN updates remain stable even in the presence of non-linearities. By treating $F(x)$ as a Lyapunov function, the positive definiteness and monotonic decrease of $F(x)$ along the trajectories of the system imply that the updates are Lyapunov stable, meaning that any small perturbations to the system will not cause instability. Consequently, the PCN is guaranteed to converge to the unique equilibrium point $x = x^*$ as training progresses.

Finally, our last theorem establishes a comparison in terms of convergence speed to fixed points between PCNs and the popular workhorse training algorithm of deep neural networks, backprop (BP). Crucially, our result formally shows that, assuming both PCNs and BP-trained networks start from the same ideal initial conditions, PCNs converge *faster* – in terms of number of needed parameter updates – to fixed points than equivalent neuronal systems trained via BP.

**Theorem 3.4.** *Let $M$ be a PCN that minimizes a free energy $F = L + \tilde{E}$, where $L$ is the backpropagation loss and $\tilde{E}$ is the residual energy. Assume the activation function $f$ and its derivatives $f'$, $f''$, and $f'''$ are Lipschitz continuous with constants $K, K', K''$, and $K'''$, respectively. Then, the PCN updates are Lyapunov stable and converge to a fixed point faster than BP updates.*

Here, in establishing that a PCN minimizer of $F$ is Lyapunov stable, the convergence behavior is analyzed under the assumption that the activation $f$ (and derivatives $f'$, $f''$, and $f'''$) is Lipschitz continuous (ensuring smooth, controlled changes in these functions). The significance of this result lies in showing that PCN dynamics can achieve a stable equilibrium point with faster convergence due to the inclusion of the residual energy $\tilde{E}$. This term captures complex dependencies between layers, providing additional regularization and smoothing to the gradient flow, further aiding to reduce oscillations and accelerating convergence (compared to BP). In interpreting $F$ as a Lyapunov function (as before, this allowed us to prove guaranteed convergence to a unique equilibrium), we see that the $\tilde{E}$ introduces a corrective mechanism, which ensures that the PCN reaches its fixed point more efficiently than BP, making it particularly useful for training deep networks where standard BP may suffer from slower convergence/instability (proof/derivations in appendix).

Taken together, the above results rigorously demonstrate, from a dynamical systems perspective, why PCNs tend to quickly converge to stable solutions as compared to BP. This rapid convergence, despite extra computational complexity incurred by an EM-like optimization, is well-supported by a body of experimental evidence Ororbia & Kifer (2022); Alonso et al. (2022); Song et al. (2023).

## 3.1 ROBUSTNESS ANALYSIS

Next, we turn to our result to formally characterize the robustness of PCNs compared to BP-adapted networks. With $L(W)$ corresponding to the BP loss and $\tilde{E}(W)$ representing the residual energy of the system, we are able to establish that a PCN's trajectory (through a loss surface) is Lyapunov stable. While BP (Backpropagation) optimizes $L(W)$ by directly minimizing the loss through gradient descent updates, PCNs take into account both $L(W)$ and $\tilde{E}(W)$, resulting in a combined energy

minimization that stabilizes the optimization path. This provides a theoretical justification for the improved robustness of PCNs, as the added residual energy $\tilde{E}(W)$ acts as a corrective term that smooths out abrupt changes in the gradient landscape, leading to a more gradual convergence.

**Theorem 3.5.** *Let $V_{PC}(W) = L(W) + \tilde{E}(W)$ be a Lyapunov function, where $L$ and $\tilde{E}$ are positive definite functions that achieve their minimum at $W = W^*$. Then, the time derivative of $V_{PC}(W)$ along the trajectories of the system, and the predictive coding updates seen as a continuous-time dynamical system are, respectively:*

$$\dot{V}_{PC}(W) = -\left\| \frac{\partial L}{\partial W} + \frac{\partial \tilde{E}}{\partial W} \right\|^2 \leq 0, \qquad \dot{W}_l = -\left( \frac{\partial L}{\partial W_l} + \frac{\partial \tilde{E}}{\partial W_l} \right), \tag{5}$$

*making the system Lyapunov stable. Furthermore, if $\dot{V}_{PC}(W) = 0$ only at $W = W^*$, then the predictive coding updates are asymptotically stable and converge to the equilibrium point $W = W^*$.*

To better understand how this relates to standard BP, let us now analyze the derivative of $V_{\text{PC}}(W)$. If the residual energy $\tilde{E}(W) \to 0$, the update rule reduces to:

$$\dot{W}_l = -\frac{\partial L}{\partial W_l},$$

which is precisely the standard BP update. Thus, BP can be seen as a special case of PCNs where the residual energy term vanishes. However, in practical scenarios, $\tilde{E}(W)$ typically does not reach zero, allowing PCNs to maintain more stable trajectories through the loss landscape. This difference becomes evident in the Lyapunov stability analysis. Since $\tilde{E}(W)$ is a positive definite function, it introduces an additional stabilizing force in the gradient dynamics, ensuring that any perturbations in $W$ are corrected by $\tilde{E}(W)$. This means that, unlike BP, which optimizes $L(W)$ alone, the combined energy minimization in PCNs results in a smoother convergence trajectory, reducing sensitivity to local minima and sharp changes in the loss surface.

Let us again consider $V_{\text{PC}}(W)$, where $L$ and $\tilde{E}$ are positive definite and achieve their minima at the equilibrium point $W = W^*$. This composite energy function serves as a Lyapunov function for the system, as shown in the above theorem. We now state a more powerful result that establishes the robustness of PCNs compared to BP.

**Theorem 3.6** (Robustness of Predictive Coding Networks). *Let $V_{PC}(W) = L(W) + \tilde{E}(W)$ be the Lyapunov function of a PCN, where $L$ and $\tilde{E}$ are positive definite functions that achieve their minima at $W = W^*$. Assume the following conditions hold:*

1. *The Hessian of $L(W)$, denoted as $H_L = \frac{\partial^2 L}{\partial W^2}$, is positive semi-definite.*
2. *The residual energy term $\tilde{E}(W)$ satisfies a Lipschitz continuity condition, i.e., there exists a constant $K > 0$ such that:*

$$\left\| \frac{\partial \tilde{E}}{\partial W} \right\| \leq K \left\| W - W^* \right\|.$$

3. *The time derivative of $V_{PC}(W)$ along the system trajectories satisfies equation 5 (left).*
4. *The initial perturbation $\Delta W$ is small, i.e., $\|\Delta W\| \ll 1$.*

*Under these conditions, the following robustness property holds for PCNs:*

Bounded Perturbation Recovery: *For any perturbation $\Delta W$ such that $W = W^* + \Delta W$, the PCN's weight trajectory $W(t)$ satisfies:*

$$\|W(t) - W^*\| \leq Ce^{-\lambda t}\|\Delta W\| + O(\epsilon),$$

*where constants $C, \lambda > 0$ depend on properties of $L(W)$ and $\tilde{E}(W)$; $\epsilon$ is perturbation magnitude. This result guarantees **exponential convergence** to $W^*$ and **robustness to bounded perturbations**.*

Effectively, these results establish that PCNs reliably converge to equilibrium points of the loss function when optimizing their variational free energy, and once they do, the parameters will remain near those points indefinitely, highlighting the robustness of PCNs.

**Robustness Comparison between PC and BP.** As one may observe above, a core difference between a BP-based network and a PCN is their underlying optimization objective – BP optimizes just $L(W)$ whereas a PCN optimizes $L(W) + \tilde{E}(W)$. The central mechanism behind a PCN's effectiveness is its use of and focus on optimizing energy $\tilde{E}$, which, as our results in Section 3 established, underwrites the PCN's ability to stably converge to fixed points (equilibrium). There are several consequences that come as a result of this crucial difference:

- *Separation Characteristic:* For BP, robustness is characterized by the minimal separation characteristic, which can be improved but remains sensitive to input deviations. However, for PCNs, robustness is higher due to stabilization provided by residual energy $\tilde{E}$.
- *Error Handling:* BP updates only consider the gradient of the loss function $L$, which makes the resulting network models more prone to overfitting and less robust to perturbations. PCN updates, in contrast, consider both $L$ and $\tilde{E}$, ensuring smoother transitions and better handling of input deviations.
- *Convergence:* BP updates can lead to oscillations or slower convergence due to their sensitivity to initial conditions and perturbations. On the other hand, PCN updates converge faster and more smoothly to equilibrium due to the damping effect of $\tilde{E}$.

### PREDICTIVE CODING UPDATES AS QUASI-NEWTON UPDATES

Quasi-Newton (QN) methods, such as BFGS optimization Head & Zerner (1985), iteratively update an approximation of the Hessian $H^{-1}$, leading to more stable, faster converging updates compared to simple gradient descent. The Newton-Raphson update rule, for example, is given as follows:

$$W_{t+1} = W_t - H^{-1}\nabla L(W_t)$$

where $H$ is the Hessian matrix of the loss function $L$. Notably, the weight updated performed by PC has been shown to approximate second-order optimization methods, specifically Gauss-Newton updates Alonso et al. (2022). In this section, we extend this result by showing first that it is the residual energy term $\tilde{E}$ that modifies the gradient similarly to what the inverse Hessian does. Second, we connect this modification to QN updates, broadening the understanding of PC within this optimization framework. Finally, we compare the computed bounds against those obtained for target propagation (TP) Bengio (2014), a popular biologically plausible learning algorithm for deep neural networks, and show that PC updates are much closer to QN updates as compared to TP, while converging faster to solutions. Let us start with the following:

**Theorem 3.7.** *Let $M$ be a neural network that minimizes a free energy $F = L + \tilde{E}$, where $L$ is the loss function and $\tilde{E}$ is the residual energy. Assume the activation function $f$ and its derivatives $f', f'', and f'''$ are Lipschitz continuous with constants $K, K', K'', and K'''$, respectively. Then, the PC updates approximate quasi-Newton updates by incorporating higher-order information through the residual energy term $\tilde{E}$.*

Given the above, let us now assume that we are faced with a learning task where we prefer using a stable training algorithm. As QN updates are well-known for their stability Head & Zerner (1985), we could favor one algorithm over another if the first better approximates such updates. Motivated by this, we next compare PC and TP, showing that the first better approximates QN updates:

**Theorem 3.8.** *Let $M$ be a neural network minimizing a free energy function $F = L + \tilde{E}$, where $L$ is the loss function and $\tilde{E}$ is a residual energy term. Let $H_{GN} = J^T J$ denote the Gauss-Newton matrix, where $J$ is the Jacobian matrix of the network's output with respect to the parameters. Consider the following update rules:*

- *QN Update: $\Delta W_{QN} = -H_{GN}^{-1}\nabla L(W)$,*
- *TP Update: $\Delta W_{TP} = -B_l^{-1}\nabla L(W)$, where $B_l$ is a block-diagonal approximation of the matrix $H_{GN}$,*
- *PC Update: $\Delta W_{PC} = -(H + \tilde{H})^{-1}\left(\nabla L(W) + \nabla \tilde{E}(W)\right)$.*

*Assume that $L(W)$ and $\tilde{E}(W)$ are twice differentiable, and the activation function $f$ and its derivatives are Lipschitz continuous. Then, the approximation error between the updates and the true Quasi-Newton update satisfies $E_{PC} \leq E_{TP}$, where:*

$$E_{PC} = \|\Delta W_{PC} - \Delta W_{QN}\|, \quad E_{TP} = \|\Delta W_{TP} - \Delta W_{QN}\|.$$

*Thus, PC is mathematically closer to the true Quasi-Newton updates than TP.*

The following theorem builds on the previous result and formally proves that PC reaches fixed points faster than TP, bolstering the value of PC bio-inspired learning over approaches such as TP.

**Theorem 3.9.** *Let $D_{PC}$ and $D_{TP}$ represent two dynamical systems corresponding to predictive coding (PC) and target propagation (TP), respectively. Assume that both systems have the same equilibrium point $W^*$, and the loss functions $L$ and residual energy $\tilde{E}$ are twice differentiable and positive definite. Furthermore, let the activation functions $f_l$ and their derivatives be Lipschitz continuous. Define the Lyapunov functions:*

1. *For predictive coding: $V_{PC}(W) = F(W) = L(W) + \tilde{E}(W)$.*
2. *For target propagation: $V_{TP}(W) = \sum_{l=1}^{L} \frac{1}{2}\|t_l - f_l(W_l t_{l-1})\|^2$.*

*Then, it follows that PC is more stable than TP according to the following criteria:*

1. *Convergence Rate: $D_{PC}$ converges faster to its equilibrium point compared to $D_{TP}$.*
2. *Deviation from Equilibrium: $D_{PC}$ exhibits a smaller deviation from the equilibrium compared to $D_{TP}$ for the same initial perturbation.*
3. *Region of Attraction: The region of attraction for $D_{PC}$ is larger than that of $D_{TP}$.*

The above highlights the fact that an important factor behind PC's successful ability to traverse to fixed-points reliably and stably is its ability to approximate the updates yielded by quasi-Newton methodology. This means that a PCN's optimization of its free energy functional affords it second-order information (without the typically prohibitive cost of computing the Hessian itself) in its traversal of a loss surface landscape. A desirable positive consequence of this section's results is that PC yields a better approximation of QN updates that the popular TP algorithm and, furthermore, PC-based learning is more stable than that of TP-centric adaptation (due to PC's ability to converge faster to equilibrium due to its smaller deviation from region of attractions in its loss surface).

## 4 SIMULATION RESULTS

In this section, we provide empirical evidence related to some of the theoretical claims made by this work. First, we test convergence bounds for PC and BP, using an MSE loss, trained on MNIST and CIFAR10. Figures 1 and 2 show a custom convolution model (2 conv layer, 3 fc layer) trained on MNIST using TeLU and tanh activations. On MNIST, we see that PC converges faster than BP, even though both approaches reach similar performance; however, the different parameter updates of each yielded different trajectories. Similar findings are seen in Figure 3 for the CIFAR10-trained model.

Next, we conduct large-scale experiments. Specifically, we train multiple models on image classification tasks using PC, BP, and TP, as well as key variations, and report the number of epochs (which is proportional to the number of weight updates) needed to reach the best performance. Our computer simulations involved testing across across four widely-used image classification datasets, offering results for both useful digit and clothing object recognition but also for the case of more complex natural images: 1) **MNIST**, 2) **Fashion-MNIST**, 3) **CIFAR10**, and 4) **CIFAR100**. The variant algorithm methods included in our study were backprop (**BP**), predictive coding (PC) that optimizes a free energy functional made up of squared error losses **PC-SE**, predictive coding that optimizes a free energy functional composed of cross-entropy losses **PC-CE**, **DDTP-Linear** Meulemans et al. (2020), **DDTP-Control** Meulemans et al. (2020), and **DTP-$\sigma$** Ororbia & Mali (2019).

Each model was trained for 100 epochs using architectures designed for the characteristics of each dataset. Additionally, beside standard activation such as Tanh, ReLU, sigmoid we incorporated a novel activation function, the hyperbolic tangent exponential linear unit (**TeLU**) Fernandez & Mali (2024), in all experiments which offered better stability. Unlike ReLU, which lacks higher-order Lipschitz continuity, TeLU, Tanh and sigmoid ensures smooth transitions and stability even

| Method | MNIST | Fashion-MNIST | CIFAR10 | CIFAR100 |
|---|---|---|---|---|
| **BP** | $98.03 \pm 0.05\%$ | $89.07 \pm 0.09\%$ | $88.16 \pm 0.17\%$ | $60.86 \pm 0.39\%$ |
| **PC-SE** | $98.4 \pm 0.05\%$ | $89.52 \pm 0.07\%$ | $87.97 \pm 0.11\%$ | $54.92 \pm 0.11\%$ |
| **PC-CE** | $98.23 \pm 0.08\%$ | $89.11 \pm 0.19\%$ | $88.02 \pm 0.14\%$ | $61.94 \pm 0.12\%$ |
| **DDTP-Linear** | $98.17 \pm 0.09\%$ | $88.99 \pm 0.17\%$ | $83.01 \pm 0.20\%$ | $57.80 \pm 0.43\%$ |
| **DDTP-control** | $98.20 \pm 0.08\%$ | $88.70 \pm 0.31\%$ | $82.71 \pm 0.28\%$ | $54.75 \pm 0.43\%$ |
| **DTP-$\sigma$** | $98.34 \pm 0.19\%$ | $89.01 \pm 0.53\%$ | $49.49 \pm 0.23\%$ | $31.74 \pm 0.30\%$ |

Table 1: Test accuracy corresponding to the epoch with the best validation error over the 100 training epochs (architecture with 5 fully-connected hidden layers of 256 units for MNIST & Fashion-MNIST; VGG-5 architecture employed for CIFAR10 and CIFAR100). Mean $\pm$ SD for 15 random seeds. Best test errors (except BP) are displayed in bold.

| Method | SGD (lr = 0.01) | SGD+M (lr = 1e-3) | RMSProp(lr = 1e-3) |
|---|---|---|---|
| **BP** | 74 | 70 | 71 |
| **PC-CE** | 62 | 63 | 65 |
| **PC-SE** | 63 | 61 | 64 |
| **DDTP-Linear** | 73 | 68 | 70 |
| **DDTP-Control** | 71 | 72 | 71 |

Table 2: Average number of epochs required to obtain best validation accuracy on Cifar-10 for various methods using different optimizers.

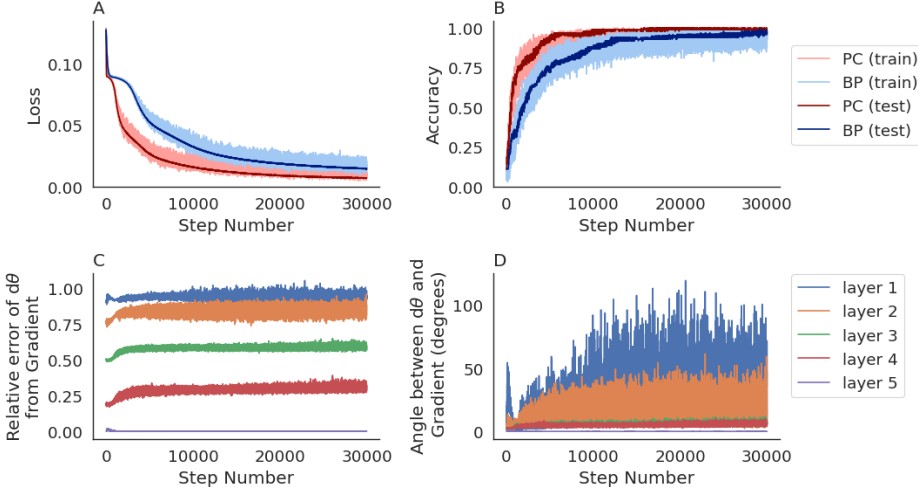

Figure 1: Convergence comparison of backprop (BP; blue) and predictive coding (PC; red) in a convolutional network with MSE loss trained on MNIST for $30K\,steps$. Panels A and B show the training (light colors) and test (dark colors) loss (A) and accuracy (B) for a 5-layer network (TeLU activation, optimizer = SGD with momentum 0.9, lr = 0.01, batch_size= 100 ) trained using PC (red) and BP (blue). Panels C and D depict the relative error (C) and angle (D) between parameter updates (d$\theta$) and the negative gradient of the loss at each layer. While PC and BP achieve comparable accuracies in all experiments, the differences in the parameter updates highlight the nuances between the two approaches. (Note: We adapted the code of Rosenbaum (2022) to generate these plots.)

for higher-order derivatives, a property crucial for this paper's theoretical findings. This property proved particularly beneficial for the predictive coding (PC) and difference target propagation (DTP) methods, resulting in more stable, consistent convergence behavior than traditional activations (using Relu/Tanh, PC exhibited higher variance.

All of the models were optimized using stochastic gradient descent (SGD) with momentum, specifically using a learning rate set to $0.001$ with a batch size $128$. For each method and dataset, we ran the experiments for 15 trials (with 15 different random seeds) and report the mean test accuracy and standard deviation (SD) corresponding to the epoch with the best validation error. The experiments were performed using *PCX*, a novel library introduced to perform experiments with and benchmark predictive coding networks Pinchetti et al. (2024). A more detailed description of the models used, as well as the test accuracies reached during the experiments, is reported in Table 4, while the the average number of epochs required for each algorithm to converge on the complex CIFAR-10 dataset

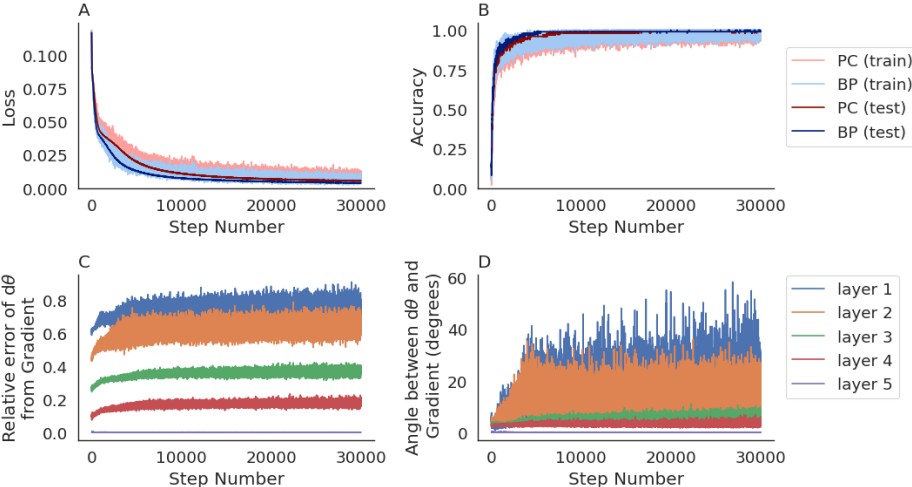

Figure 2: Convergence analysis with the tanh activation – same setting as in Figure 1.

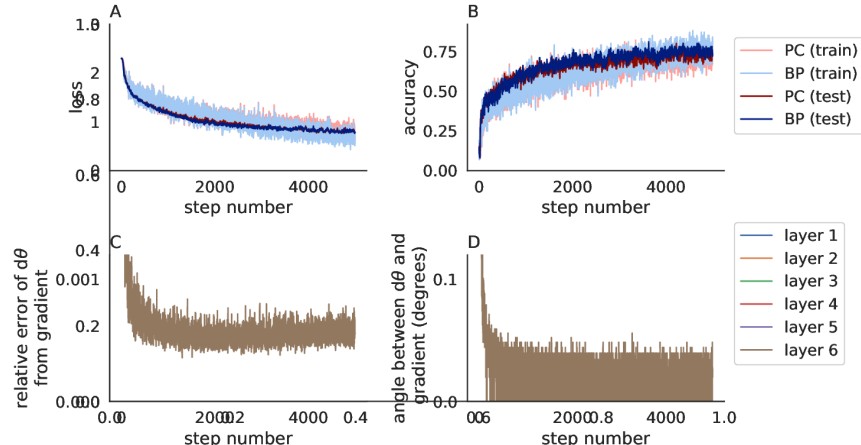

Figure 3: Convergence comparison of backprop (BP; blue) and predictive coding (PC; red) in a convolutional network with MSE loss trained on the CIFAR-10 dataset over first $4K$ steps. Panels A and B show the training (light colors) and test (dark colors) loss (A) and accuracy (B) for a 5-layer network (TeLU activation, optimizer = SGD with momentum 0.9, lr = 0.01, batch_size= 100 ) trained using PC shown in red) and BP (shown in blue). Panels C and D depict the relative error (C) and angle (D) between the parameter updates,(d$\theta$) and the negative gradient of the loss at each layer

is reported in Table 2. Note that, in Table 2, we investigated each algorithm used in tandem with either SGD, SGD with momentum (SGD+M), or RMSprop Tieleman et al. (2012).

## 5 CONCLUSION

In this work, we conducted a rigorous study of the stability and robustness of the biological inference and learning framework known as predictive coding (PC). Concretely, we employed dynamical systems theory to show that PC networks (PCNs) are Lyapunov stable by specifically deriving tights bounds on the stability of PCNs equipped with Lipschitz activation functions, demonstrating that they are able to converge to fixed-points and further offer robustness to perturbations applied to sensory inputs, synaptic weight values, and initial neuronal state conditions. Furthermore, we were able to formally demonstrate that the updates yielded by PCN dynamics approximate quasi-Newton updates applied to synaptic connection strengths, uncovering that PCNs utilize second order information to adjust the direction and magnitude of the gradient steps that they carry out. Notably, our results usefully yielded tighter theoretical bounds on the approximations of other algorithms, such as backpropagation of errors and target propagation, that are yielded by predictive coding models.

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

In this appendix, we provide additional experimental details as well as the complete formalization/set of details underlying the definitions and theoretical results provided in this work, including the formal proofs behind the presented theorems.

## A    EXPERIMENTAL DETAILS

In this section, in Table 3 (Left), we provide additional details with respect to the full specification of the model architectures utilized for our credit assignment algorithms in the main paper's simulations. Furthermore, we present hyper-parameter value settings used for each credit assignment algorithm in Table 3 (Right). Hyper-parameter search where conducted over optimizer(SGD, SGD+M, AdamW), learning rate ([0.1, 1e-4]) and activation ([Tanh, Sigmoid, TeLU, ReLU]).

| Model | Architecture |
|---|---|
| MNIST | $5 \times 256$ FC, Glorot-uniform init |
| FMNIST | $5 \times 256$ FC, Glorot-uniform init |
| CIFAR10 | VGG5, Glorot-uniform init |
| CIFAR100 | VGG5, Glorot-uniform init |
| Algorithm | Hyper-parameter Configuration |
| BP | SGD, lr=1e-3, Momentum= 0.9, Batch_size=128 |
| DTP-Linear | SGD, lr=1e-4, Momentum= 0.9, Batch_size=128 |
| DTP-Control | SGD, lr=1e-4, Momentum= 0.92, Batch_size=128 |
| DTP-$\sigma$ | SGD, lr=1e-2, Momentum= 0.91, Batch_size=128 |
| PC-SE | SGD, lr=1e-3, Momentum= 0.9, Batch_size=128 |
| PC-CE | SGD, lr=1e-3, Momentum= 0.9, Batch_size=128 |

Table 3: (Left) Architecture details used for the neural models simulated in this work. (Right) Algorithm hyper-parameter configuration for each credit assignment scheme explored in the main paper. 'FC' denotes fully-connected, 'SGD' denotes stochastic gradient descent, and VGG5 refers to the convolution vision architecture designed in Simonyan & Zisserman (2014). Note that all parameter/synaptic weight values were initialized from a fan-in-scaled initialization Glorot & Bengio (2010) scheme ("Glorot-uniform init"), except for the biases, which were initialized to zeros.

## B    DEFINITIONS

In this section, we fully formalize the background section on dynamical systems, made more informal due to a lack of space. The theorems stated in the main body of the paper, and proved in the following sections, rely on the following definitions. We start by first characterizing what it means for a dynamic system to be *stable*. This is established by the following definition:

**Definition B.1.** *A dynamical system is said to be* stable *if, for any given $\epsilon > 0$, there exists a $\delta > 0$ such that if the initial state $x(0)$ is within $\delta$ of an equilibrium point $x^*$ (i.e., $\|x(0) - x^*\| < \delta$), then the state $x(t)$ remains within $\epsilon$ of $x^*$ for all $t \geq 0$ (i.e., $\|x(t) - x^*\| < \epsilon$ for all $t \geq 0$).*

*Formally:*

$$\forall \epsilon > 0, \exists \delta > 0 \text{ such that } \|x(0) - x^*\| < \delta \implies \|x(t) - x^*\| < \epsilon \text{ for all } t \geq 0.$$

Next we compare stability between two dynamical systems and show conditions required to show one system is better compared to another, which is formally defined as follows:

**Definition B.2.** *Let $D$ and $D'$ be two dynamical systems with equilibrium points $x_D^*$ and $x_{D'}^*$, respectively. We say that the dynamical system $D$ is* more stable *than the dynamical system $D'$ if, for the same perturbation or initial deviation from the equilibrium, the system $D$ either:*

   *(i)  Converges to its equilibrium point faster than $D'$, i.e., it has a higher convergence rate, or*

   *(ii) Remains closer to its equilibrium point for all time, i.e., has a smaller deviation from the equilibrium for the same perturbation, or*

   *(iii) Has a larger region of attraction, indicating a greater tolerance to perturbations.*

*Formally, consider two dynamical systems $D$ and $D'$ represented by the differential equations:*

$$\dot{x}_D = f_D(x), \quad \dot{x}_{D'} = f_{D'}(x),$$

*with equilibrium points $x_D^*$ and $x_{D'}^*$, respectively. We define three criteria:*

*1. Convergence Rate: $D$ converges faster to $x_D^*$ compared to $D'$ if there exists a constant $\lambda > 0$ such that, for initial conditions $x(0)$ close to $x_D^*$ and $x(0)$ close to $x_{D'}^*$:*

$$\|x_D(t) - x_D^*\| \leq e^{-\lambda t}\|x(0) - x_D^*\|, \quad \|x_{D'}(t) - x_{D'}^*\| \geq e^{-\lambda' t}\|x(0) - x_{D'}^*\|, \quad \text{with} \quad \lambda > \lambda'.$$

*2. Region of Attraction: $D$ has a larger region of attraction if the set of initial points that converge to $x_D^*$, denoted as $\mathcal{R}_D$, satisfies:*

$$\mathcal{R}_D \supseteq \mathcal{R}_{D'}.$$

*3. Deviation from Equilibrium: For the same initial deviation $\|x(0) - x_D^*\| = \|x(0) - x_{D'}^*\|$, the system $D$ satisfies:*

$$\|x_D(t) - x_D^*\| \leq \|x_{D'}(t) - x_{D'}^*\|, \quad \forall t \geq 0.$$

*If any of these criteria are met, then we say that $D$ is* more stable *than $D'$.*

Next we show important property of a system reaching fixed point, which is formally defined as follows:

**Definition B.3.** *Let $x(t) \in \mathbb{R}^n$ denote the state of a dynamical system at time $t$, governed by the differential equation:*

$$\dot{x}(t) = f(x(t)),$$

*with $f : \mathbb{R}^n \to \mathbb{R}^n$ a continuous function. A point $x^* \in \mathbb{R}^n$ is called a* fixed point *or* equilibrium point *of the system if $f(x^*) = 0$. The system is said to* converge to the fixed point $x^*$ *if, for any initial state $x(0)$ in a neighborhood $\mathcal{N}$ of $x^*$, the trajectory $x(t)$ satisfies:*

$$\lim_{t \to \infty} x(t) = x^*.$$

*More formally, for a neighborhood $\mathcal{N} \subseteq \mathbb{R}^n$ of $x^*$, there exists a constant $\delta > 0$ such that if $\|x(0) - x^*\| < \delta$, then:*

$$\lim_{t \to \infty} x(t) = x^*.$$

*If the above holds for every initial condition $x(0) \in \mathbb{R}^n$, then $x^*$ is said to be a* globally stable fixed point.

Next we formally define rate of convergence between two stable dynamical systems and how can be used to measure convergence bound for a network, which is defined as follows:

**Definition B.4.** *Let $D$ and $D'$ be dynamical systems that converge to the same fixed point $x^*$ given the same initial conditions. Let $x_D(t)$ and $x_{D'}(t)$ denote the trajectories of the systems $D$ and $D'$, respectively, starting from the same initial state $x(0)$. We say that $D$ converges* faster *than $D'$ if, for any initial state $x(0)$ sufficiently close to $x^*$, the distance to the fixed point decreases at a higher rate for $D$ compared to $D'$.*

*Formally, $D$ converges faster than $D'$ if there exists a positive constant $\lambda > 0$ such that for all $t \geq 0$:*

$$\|x_D(t) - x^*\| \leq Ce^{-\lambda t}\|x(0) - x^*\|, \quad \text{and} \quad \|x_{D'}(t) - x^*\| \geq C'e^{-\lambda' t}\|x(0) - x^*\|, \quad \text{with} \quad \lambda > \lambda',$$

*where $C, C' > 0$ are positive constants and $\lambda, \lambda'$ represent the exponential convergence rates of the systems $D$ and $D'$, respectively.*

*Alternatively, for non-exponential convergence, $D$ converges faster than $D'$ if, for any $\epsilon > 0$, there exist times $T_D, T_{D'} > 0$ such that:*

$$T_D \leq T_{D'} \quad \text{and} \quad \|x_D(t) - x^*\| < \epsilon \quad \text{for all } t \geq T_D, \quad \|x_{D'}(t) - x^*\| < \epsilon \quad \text{for all } t \geq T_{D'}.$$

Next we formally define robustness of a dynamical system

**Definition B.5.** *A dynamical system is said to be* robust to random perturbations *if, for any small random perturbation $\eta(t)$ acting on the system, the perturbed state $x_\eta(t)$ remains close to the unperturbed state $x^*(t)$ with high probability for all $t \geq 0$. This implies that the system can maintain its desired performance despite the presence of small random disturbances.*

*Formally, let $x(t)$ be the unperturbed state and $x_\eta(t)$ be the perturbed state of the system. The system is robust if, for any $\epsilon > 0$ and any initial condition $x(0)$, there exists a constant $\delta > 0$ and a probability threshold $1 - \alpha$, such that if $\|\eta(t)\| < \delta$ for all $t \geq 0$, then:*

$$\mathbb{P}\left(\sup_{t \geq 0} \|x_\eta(t) - x(t)\| < \epsilon\right) \geq 1 - \alpha.$$

Next now we formally define what it means for a dynamical system to be robust to initial conditions.

**Definition B.6.** *A dynamical system is said to be* robust to random initial conditions *if, given an initial state $x(0)$ drawn from a probability distribution $\mathcal{P}$ with bounded support, the system's state $x(t)$ remains within a bounded region around its desired trajectory or equilibrium point $x^*(t)$ with high probability for all $t \geq 0$.*

*Formally, let the initial state $x(0)$ be drawn from a distribution $\mathcal{P}$ with support $S$ such that $\|x(0) - x^*(0)\| \leq M$ for some $M > 0$. The system is robust if there exists a probability threshold $1 - \alpha$ such that for any $\epsilon > 0$, there exists a constant $\delta > 0$ satisfying:*

$$\mathbb{P}\left(\sup_{t \geq 0} \|x(t) - x^*(t)\| < \epsilon\right) \geq 1 - \alpha \quad \text{whenever} \quad \|x(0) - x^*(0)\| \leq \delta.$$

**Definition B.7.** *A neural network is said to be* Lyapunov stable *if, for any given $\epsilon > 0$, there exists a $\delta > 0$ such that if the initial weights $W(0)$ are within $\delta$ of a trained equilibrium point $W^*$ (i.e., $\|W(0) - W^*\| < \delta$), then the weights $W(t)$ remain within $\epsilon$ of $W^*$ for all $t \geq 0$ (i.e., $\|W(t) - W^*\| < \epsilon$ for all $t \geq 0$).*

*If, in addition, the weights converge to $W^*$ as $t \to \infty$, i.e.,*

$$\lim_{t \to \infty} \|W(t) - W^*\| = 0,$$

*then the neural network is said to be* asymptotically stable.

**Example 1.** Why is Stability Important?

*Stability is a fundamental property of neural networks, ensuring that small changes in the input or initial conditions do not cause large, unpredictable variations in the network's output. Stability is crucial for both training and deployment. During training, instability can cause gradients to explode, leading to chaotic updates and preventing convergence. During deployment, an unstable network might produce drastically different outputs for minor input changes, undermining its reliability. In real-world scenarios where inputs are often noisy or slightly perturbed, a stable network can still produce accurate and consistent predictions, making it more reliable and robust.*

Mathematical Insight:

*Stability can be formally understood through the concept of \*bounded sensitivity\*. For a neural network $f : \mathbb{R}^n \to \mathbb{R}^m$ to be stable, a small change in the input $\delta x$ should lead to a bounded change in the output. Mathematically, this is expressed as:*

$$\|f(x + \delta x) - f(x)\| \leq K \cdot \|\delta x\|,$$

*for a constant $K > 0$. A network is \*stable\* if $K$ is small, implying that the network is not highly sensitive to input variations. A more refined way to analyze stability is through the eigenvalues of the Jacobian matrix $J_f(x)$. If the largest eigenvalue $\lambda_{\max}$ of $J_f(x)$ satisfies $\lambda_{\max} < 1$, then the system is considered to be stable because small perturbations in the input will decay rather than amplify as they propagate through the network.*

Intuitive Example of Stability:

*Consider a neural network trained for handwritten digit classification. If the network is stable, adding a small amount of Gaussian noise or slightly changing the position of a digit in the image*

*(e.g., shifting the digit by a few pixels) will not cause the network's prediction to change significantly. For example, a stable network will continue to classify a digit '8' as '8', even if the digit is slightly blurred or has a few pixels altered. This stability is essential for real-world applications where inputs are rarely perfect replicas of training data.*

Detailed Example: Training a Recurrent Neural Network (RNN)

*Stability becomes even more critical in Recurrent Neural Networks (RNNs) due to their sequential nature. Consider an RNN designed to predict the next word in a sentence. During training, if small changes in initial states or earlier inputs cause large deviations in hidden states, the RNN can produce completely different outputs, making training highly unstable. Mathematically, this corresponds to having a Jacobian matrix with eigenvalues greater than 1, which indicates that small errors will grow exponentially as they propagate through the network.*

Counterexample: Non-Stable Systems and Vanishing/Exploding Gradients

*Imagine training a deep feedforward network or an RNN without appropriate regularization. If the network is not stable, even small changes in the input or initial weights can lead to vanishing or exploding gradients, making learning ineffective. For example, in a deep RNN, an input sequence with slightly perturbed values might cause the gradients to either shrink to zero (vanishing gradient problem) or explode to very large values (exploding gradient problem), leading to poor model performance and unstable training.*

Practical Implications of Stability:

*1. Improved Training Dynamics: Stability ensures that gradient-based optimization methods produce smooth and predictable updates, enabling faster and more reliable convergence. 2. Reliable Predictions: In deployment, stable networks produce consistent outputs even with noisy or perturbed inputs, making them suitable for real-world applications like speech recognition or autonomous driving. 3. Control Over Adversarial Attacks: A stable network is less vulnerable to adversarial attacks, where small, carefully crafted input changes are designed to mislead the network into producing incorrect outputs.*

Real-World Example: Stability in Financial Forecasting

*Consider a neural network used for predicting stock prices. Financial data is inherently noisy and contains sudden fluctuations. If the network is unstable, even a slight change in the input features (e.g., a minor fluctuation in daily stock prices) could lead to drastically different predictions, making the network unreliable for decision-making. On the other hand, a stable network will produce consistent and reliable forecasts, ensuring that minor changes in the data do not cause significant shifts in the predictions.*

**Definition B.8.** *A neural network is said to be* robust to small random perturbations *if, for any small random perturbation $\eta(t)$ applied to its inputs or parameters, the network's output $y_\eta(t)$ remains close to its unperturbed output $y^*(t)$ with high probability for all $t \geq 0$.*

*Formally, let $y(t)$ be the nominal (unperturbed) output of the neural network and $y_\eta(t)$ be the output under random perturbations $\eta(t)$. The network is said to be robust if, for any $\epsilon > 0$, there exists a constant $\delta > 0$ and a probability threshold $1 - \alpha$, such that if $\|\eta(t)\| < \delta$ for all $t \geq 0$, then:*

$$\mathbb{P}\left(\sup_{t \geq 0} \|y_\eta(t) - y^*(t)\| < \epsilon\right) \geq 1 - \alpha.$$

**Example 2.** Why is Robustness Important?

*Robustness is a critical property for neural networks, especially in real-world applications, where input data is often noisy, contains slight perturbations, or deviates from the clean training data. A robust neural network maintains its accuracy and reliability even when faced with unexpected changes, ensuring consistent performance across diverse and challenging conditions. Robustness is crucial for safety-critical systems (e.g., autonomous vehicles, medical diagnosis), as failures can have significant real-world consequences.*

Mathematical Intuition:

*Robustness can be understood mathematically through the concept of \*Lipschitz continuity\*. For a neural network $f$ to be robust, small changes in the input should only cause small changes in the output. Formally, if $f$ is $K$-Lipschitz continuous, then for any two inputs $x$ and $x'$:*

$$\|f(x) - f(x')\| \le K \cdot \|x - x'\|,$$

*where $K$ is a constant that quantifies the network's sensitivity to input variations. A small $K$ implies that the network is less sensitive to input noise, thereby enhancing robustness. Robust networks minimize $K$ and ensure that even adversarial or noisy inputs do not cause large deviations in the output, making them more reliable.*

Intuitive Example of Robustness in Autonomous Driving:

*Consider a neural network used for object detection in an autonomous vehicle. In ideal conditions, the network correctly identifies cars, pedestrians, and traffic signs. However, in real-world scenarios, the network must operate in varying lighting conditions (e.g., night vs. day), different weather (e.g., fog, rain), and even with potential sensor noise (e.g., camera blur, reflections).*

*If the network is robust, a slight distortion in a camera image—such as a shadow, slight rain droplets, or fog—will not cause it to misclassify a pedestrian as a signpost or ignore a stop sign. The network's ability to generalize well in the presence of such variations ensures safety and reliability.*

Detailed Example: Image Classification under Adversarial Noise

*Let's consider a neural network trained to classify handwritten digits from the MNIST dataset. If the network is only trained on clean images, adding small perturbations (e.g., random Gaussian noise) might lead to misclassification, even though a human observer would still recognize the digit.*

*To test robustness, we introduce small adversarial noise $\delta$ to the input images such that the perturbed image is $x' = x + \delta$. For a non-robust network, a minimal $\delta$ can cause the classification result to change drastically, even though $\|\delta\|$ is small. A robust network, however, would maintain consistent outputs for both $x$ and $x'$, implying that $f(x') \approx f(x)$.*

Counterexample: Non-Robust Systems in Medical Imaging

*Consider a neural network designed to detect tumors in medical images. If the network is not robust, even slight variations in the input—such as minor differences in lighting or slight changes in the position of the patient—could lead to incorrect classifications, potentially causing false positives or missing a critical diagnosis. In medical applications, where errors have life-threatening consequences, robustness is paramount.*

Real-World Consequences of Non-Robustness

*In safety-critical applications like autonomous driving or healthcare, a non-robust network can have catastrophic outcomes. For example, an autonomous vehicle that fails to detect a pedestrian due to a shadow or slight occlusion might cause an accident. Similarly, a non-robust medical diagnostic system could misinterpret scans due to slight noise, leading to misdiagnosis.*

**Definition B.9.** *Let $N$ and $N'$ be two neural networks with weight dynamics governed by the differential equations:*

$$\dot{W}_N(t) = F_N(W_N(t)), \quad \dot{W}_{N'}(t) = F_{N'}(W_{N'}(t)),$$

*where $W_N(t)$ and $W_{N'}(t)$ represent the weights of the networks $N$ and $N'$ at time $t$, respectively, and $W^*$ is the common equilibrium point such that $F_N(W^*) = 0$ and $F_{N'}(W^*) = 0$.*

*We say that the network $N$ converges faster than the network $N'$ if, for any initial weights $W_N(0)$ and $W_{N'}(0)$ sufficiently close to $W^*$, there exists a positive constant $\lambda > 0$ such that the following inequality holds:*

$$\|W_N(t)-W^*\| \le Ce^{-\lambda t}\|W_N(0)-W^*\|, \quad \|W_{N'}(t)-W^*\| \ge C'e^{-\lambda' t}\|W_{N'}(0)-W^*\|, \quad \text{with} \quad \lambda > \lambda',$$

*where $C, C' > 0$ are positive constants and $\lambda, \lambda'$ represent the exponential convergence rates of the systems $N$ and $N'$, respectively.*

*If the convergence is non-exponential, $N$ is said to converge faster than $N'$ if, for any $\epsilon > 0$, there exist times $T_N, T_{N'} > 0$ such that:*

$$T_N \le T_{N'} \quad and \quad \|W_N(t)-W^*\| < \epsilon \quad for\ all\ t \ge T_N, \quad \|W_{N'}(t)-W^*\| < \epsilon \quad for\ all\ t \ge T_{N'}.$$

**Example 3.** Why is Fast Convergence Important?

*Fast convergence is essential for neural networks because it significantly reduces the time and computational resources required to train complex models. In modern machine learning, where models often have millions or even billions of parameters, slow convergence can make training impractical or prohibitively expensive. Moreover, achieving faster convergence often means that the model can reach higher accuracy or better generalization in less time, making it more competitive in real-world applications. Fast convergence is particularly important for:*

- *Large-scale datasets: Reducing convergence time on massive datasets (e.g., ImageNet, OpenWebText) can translate to days or even weeks saved during training.*

- *Iterative model development: Faster convergence allows for rapid prototyping and testing, enabling researchers to experiment with more architectures and hyperparameters within the same timeframe.*

- *Dynamic or real-time systems: In applications where models need to be frequently retrained (e.g., reinforcement learning in robotics, online learning in trading systems), fast convergence is a necessity rather than a luxury.*

Mathematical Insight into Convergence Speed:

*The convergence rate of an optimization algorithm is typically characterized by the decrease in the loss function $L$ over iterations. Formally, if $L_t$ denotes the loss at iteration $t$, then the convergence rate is defined as:*

$$L_t - L^* \leq C \cdot e^{-\rho t} \quad \text{for some constants } C, \rho > 0,$$

*where $L^*$ is the optimal loss and $\rho$ is the convergence rate. A higher $\rho$ indicates faster convergence. Consider two algorithms:*

*1. Algorithm A with a high convergence rate $\rho_A$, such that $L_t - L^* \leq C \cdot e^{-\rho_A t}$.*

*2. Algorithm B with a lower convergence rate $\rho_B < \rho_A$.*

*Even if both algorithms have the same starting point and are optimizing the same loss, Algorithm A will reach near-optimal performance exponentially faster than Algorithm B. This difference becomes dramatic when $t$ is large, making Algorithm A much more suitable for training deep neural networks.*

Detailed Example: Training a Convolutional Neural Network (CNN)

*Consider a Convolutional Neural Network (CNN) designed to classify images in the CIFAR-10 dataset. Let's analyze two scenarios:*

*1. Scenario 1: We use a standard Stochastic Gradient Descent (SGD) optimizer with a constant learning rate, which is known for slow convergence on non-convex functions. As a result, the network might require 300 epochs to reach an accuracy of 85*

*2. Scenario 2: We switch to a momentum-based variant of SGD or the Adam optimizer, both of which have adaptive learning rates and gradient scaling. This change allows the same CNN to reach 85*

*The significant difference in training time between these two scenarios can be quantified as follows: if each epoch takes 5 minutes to complete, Scenario 1 would take 25 hours, while Scenario 2 would take just over 8 hours. This not only saves time but also reduces computational costs, which is crucial for large-scale deployments.*

Counterexample: Slow Convergence and Its Implications

*Let's consider training a Transformer model for language modeling on a large corpus. Suppose that due to poor hyperparameter tuning (e.g., a suboptimal learning rate schedule), the convergence rate is so slow that it requires $10x$ more epochs to reach comparable performance. The consequences are as follows:*

- *Higher computational cost: Training time may increase from 2 days to 20 days on a multi-GPU setup, inflating hardware costs.*

- *Delayed deployment: In dynamic systems, delayed training means delayed responses to changes in the data distribution, making the model outdated by the time it is deployed.*
- *Suboptimal performance: Slow convergence can sometimes lead to getting stuck in local minima or saddle points, resulting in a final model that performs worse than one trained with a faster-converging algorithm.*

Real-World Example: Fine-Tuning a BERT Model

*Consider fine-tuning a BERT model for a sentiment analysis task. Due to the size of the model and the complexity of the task, convergence can be slow. However, by using learning rate scheduling strategies such as a cosine decay or warm-up phase, we can speed up convergence significantly. Instead of taking 5 epochs to achieve 90*

*In practice, this difference translates to reduced training costs and faster model deployment. More-over, faster convergence reduces the risk of overfitting, since the model spends less time oscillating around local minima.*

*Thus faster convergence in neural networks is not just a matter of training speed—it impacts every stage of model development, from experimentation and prototyping to deployment and long-term maintenance. Thus, selecting optimizers and training strategies that promote fast convergence is crucial for efficient and effective deep learning.*

## C   THEOREMS ON THE STABILITY OF PC

### C.1   PROOF OF THEOREM 3.1

**Theorem C.1.** *Let $M$ be a Predictive Coding Network (PCN) that minimizes a free energy $F = L + \tilde{E}$, where $L$ is the backpropagation loss and $\tilde{E}$ is the residual energy. Assume the activation function $f$ and its derivatives $f'$, $f''$, and $f'''$ are Lipschitz continuous with constants $K$, $K'$, $K''$, and $K'''$, respectively. Then, the convergence and stability of the PCN can be characterized by the bounds involving these higher-order derivatives.*

*Proof.* Consider the prediction error dynamics for the $l$-th layer in a PCN, defined as:

$$\epsilon_l(t) = x_l(t) - f(W_l x_{l-1}(t)).$$

We perform a Taylor series expansion of $f(W_l x_{l-1}(t))$ around the equilibrium point $W_l x_{l-1}^*$:

$$f(W_l x_{l-1}) = f(W_l x_{l-1}^*) + f'(W_l x_{l-1}^*)(W_l(x_{l-1} - x_{l-1}^*)) + \frac{1}{2}f''(W_l x_{l-1}^*)(W_l(x_{l-1} - x_{l-1}^*))^2$$

$$+ \frac{1}{6}f'''(W_l x_{l-1}^*)(W_l(x_{l-1} - x_{l-1}^*))^3 + O((W_l(x_{l-1} - x_{l-1}^*))^4).$$

Define the residual terms:

$$R_2 = \frac{1}{2}f''(W_l x_{l-1}^*)(W_l(x_{l-1} - x_{l-1}^*))^2, \quad R_3 = \frac{1}{6}f'''(W_l x_{l-1}^*)(W_l(x_{l-1} - x_{l-1}^*))^3.$$

The prediction error can then be expressed as:

$$\epsilon_l(t) = x_l(t) - f(W_l x_{l-1}^*) - f'(W_l x_{l-1}^*)(W_l(x_{l-1} - x_{l-1}^*)) - R_2 - R_3.$$

By the Lipschitz continuity assumptions, we have the following bound for $\epsilon_l(t)$:

$$\|\epsilon_l(t)\| \le K\|x_l - f(W_l x_{l-1}^*)\| + K'\|W_l(x_{l-1} - x_{l-1}^*)\| + K''\|(W_l(x_{l-1} - x_{l-1}^*))^2\| + K'''\|(W_l(x_{l-1} - x_{l-1}^*))^3\|.$$

This completes the proof. $\square$

## C.2 Proof of Theorem 3.2

**Theorem C.2.** *Let $M$ be a Predictive Coding Network (PCN) that minimizes a free energy function $F = L + \tilde{E}$, where $L$ is the backpropagation loss and $\tilde{E}$ is the residual energy. Assume that $L(x)$ and $\tilde{E}(x)$ are positive definite, and their sum $F(x)$ has a strict minimum at $x = x^*$, where $F(x^*) = 0$. Further, assume the activation function $f$ and its derivatives $f'$, $f''$ are Lipschitz continuous with constants $K, K', K''$, respectively. Then, the PCN dynamics can be represented as a continuous-time dynamical system, and the Lyapunov function $V(x) = F(x)$ ensures convergence to the equilibrium $x = x^*$.*

*Proof.* Consider the prediction error dynamics in a PCN given by:

$$\epsilon_l = x_l - f(W_l x_{l-1}).$$

The objective of the PCN is to minimize the free energy function:

$$F(x) = L(x) + \tilde{E}(x),$$

where $L(x)$ is the backpropagation loss and $\tilde{E}(x)$ is the residual energy. To analyze the stability of this system, we introduce a Lyapunov function:

$$V(x) = F(x) = L(x) + \tilde{E}(x).$$

**Step 1: Verifying Positive Definiteness of $V(x)$**

For $V(x) = L(x) + \tilde{E}(x)$ to be positive definite, the following must hold:

1. $V(x) \geq 0$ for all $x$. 2. $V(x) > 0$ for $x \neq x^*$. 3. $V(x^*) = 0$.

We assume that both $L(x)$ and $\tilde{E}(x)$ are positive definite around $x = x^*$. This implies:

$$L(x) > 0 \quad \text{and} \quad \tilde{E}(x) > 0 \quad \text{for} \quad x \neq x^*,$$

and

$$L(x^*) = 0, \quad \tilde{E}(x^*) = 0.$$

Therefore, $V(x) = L(x) + \tilde{E}(x)$ satisfies $V(x) > 0$ for all $x \neq x^*$ and $V(x^*) = 0$. Thus, $V(x)$ is positive definite.

**Step 2: Time Derivative of $V(x)$**

The time derivative of $V(x)$ along the trajectories of the system is given by:

$$\dot{V}(x) = \frac{\partial V}{\partial x} \cdot \dot{x}.$$

Using the dynamical system equation:

$$\dot{x}_l = -\frac{\partial F}{\partial x_l} = -\left( \frac{\partial L}{\partial x_l} + \frac{\partial \tilde{E}}{\partial x_l} \right),$$

we have:

$$\dot{V}(x) = \frac{\partial F}{\partial x} \cdot \left( -\frac{\partial F}{\partial x} \right) = -\left\| \frac{\partial F}{\partial x} \right\|^2.$$

Since $\dot{V}(x) = -\left\| \frac{\partial F}{\partial x} \right\|^2 \leq 0$, $V(x)$ is non-increasing along the trajectories of the system.

**Step 3: Stability Analysis**

Because $V(x)$ is positive definite and $\dot{V}(x) = -\left\| \frac{\partial F}{\partial x} \right\|^2 \leq 0$, $V(x)$ decreases monotonically and reaches its minimum value at $x = x^*$. This implies that the system is Lyapunov stable. Furthermore, $\dot{V}(x) = 0$ if and only if $\frac{\partial F}{\partial x} = 0$, which occurs at the equilibrium point $x = x^*$.

Hence, the system converges to the equilibrium point $x = x^*$, completing the proof. $\square$

### C.3 PROOF OF THEOREM 3.3

**Theorem C.3.** *Let $M$ be a Predictive Coding Network (PCN) that minimizes a free energy function $F = L + \tilde{E}$, where $L$ is the backpropagation loss and $\tilde{E}$ is the residual energy. Assume that $L(x)$ and $\tilde{E}(x)$ are positive definite functions, and their sum $F(x)$ has a strict minimum at $x = x^*$. Further, assume the activation function $f$ and its derivatives $f'$, $f''$, and $f'''$ are Lipschitz continuous with constants $K$, $K'$, $K''$, and $K'''$, respectively. Then, the PCN updates are Lyapunov stable and converge to a fixed point $x = x^*$.*

*Proof.* Consider the prediction error dynamics in a PCN defined as:

$$\epsilon_l = x_l - f(W_l x_{l-1}).$$

The objective of the PCN is to minimize the free energy function:

$$F(x) = L(x) + \tilde{E}(x),$$

where $L(x)$ is the backpropagation loss and $\tilde{E}(x)$ is the residual energy. To analyze the stability of this system, we introduce a Lyapunov function:

$$V(x) = F(x) = L(x) + \tilde{E}(x).$$

**Step 1: Verifying Positive Definiteness of $V(x)$**

For $V(x) = L(x) + \tilde{E}(x)$ to be a valid Lyapunov function, it must be positive definite, which requires:

$$V(x) > 0 \quad \text{for all } x \neq x^* \quad \text{and} \quad V(x^*) = 0.$$

We assume that both $L(x)$ and $\tilde{E}(x)$ are positive definite around $x = x^*$. This means:

$$L(x) > 0 \quad \text{and} \quad \tilde{E}(x) > 0 \quad \text{for } x \neq x^*,$$

and

$$L(x^*) = 0, \quad \tilde{E}(x^*) = 0.$$

Thus, $V(x) = L(x) + \tilde{E}(x)$ satisfies $V(x) > 0$ for all $x \neq x^*$ and $V(x^*) = 0$.

**Step 2: Time Derivative of $V(x)$**

The time derivative of $V(x)$ along the trajectories of the system is given by:

$$\dot{V}(x) = \frac{\partial V}{\partial x} \cdot \dot{x}.$$

Using the dynamical system equation:

$$\dot{x}_l = -\frac{\partial F}{\partial x_l} = -\left( \frac{\partial L}{\partial x_l} + \frac{\partial \tilde{E}}{\partial x_l} \right),$$

we obtain:

$$\dot{V}(x) = \frac{\partial F}{\partial x} \cdot \left( -\frac{\partial F}{\partial x} \right) = -\left\| \frac{\partial F}{\partial x} \right\|^2.$$

Since $\dot{V}(x) = -\left\| \frac{\partial F}{\partial x} \right\|^2 \leq 0$, $V(x)$ is non-increasing along the trajectories of the system. This implies that the free energy $F$ decreases monotonically until the system reaches the equilibrium point $x = x^*$, where $\frac{\partial F}{\partial x} = 0$.

**Step 3: Establishing Stability and Convergence**

Because $V(x)$ is positive definite and $\dot{V}(x) = -\left\| \frac{\partial F}{\partial x} \right\|^2 \leq 0$, $V(x)$ decreases monotonically and reaches its minimum value at $x = x^*$. This implies that the system is Lyapunov stable.

Furthermore, $\dot{V}(x) = 0$ if and only if $\frac{\partial F}{\partial x} = 0$, which occurs at the equilibrium point $x = x^*$.

Thus, as $t \to \infty$, $\dot{V}(x) \to 0$ and the system converges to the equilibrium point $x = x^*$, completing the proof. $\square$

## C.4 PROOF OF THEOREM 3.4

**Theorem C.4.** *Let $M$ be a Predictive Coding Network (PCN) that minimizes a free energy $F = L + \tilde{E}$, where $L$ is the backpropagation loss and $\tilde{E}$ is the residual energy. Assume the activation function $f$ and its derivatives $f'$, $f''$, and $f'''$ are Lipschitz continuous with constants $K, K', K''$, and $K'''$, respectively. Then, the PCN updates are Lyapunov stable and converge to a fixed point faster than BP updates.*

*Proof.* **Stability Analysis of BP Updates.** Consider the BP updates modeled as a continuous-time dynamical system:

$$\dot{W}_l = -\frac{\partial L}{\partial W_l}.$$

Define a Lyapunov function for the BP updates:

$$V_{\text{BP}}(W) = L(W).$$

The time derivative of $V_{\text{BP}}$ along the trajectories of the system is:

$$\dot{V}_{\text{BP}}(W) = \frac{\partial V_{\text{BP}}}{\partial W} \cdot \dot{W} = -\left(\frac{\partial L}{\partial W}\right)^T \cdot \frac{\partial L}{\partial W} = -\left\|\frac{\partial L}{\partial W}\right\|^2.$$

Since $\dot{V}_{\text{BP}} \leq 0$, $V_{\text{BP}}$ is non-increasing along the trajectories of the BP updates. This indicates that the BP updates are Lyapunov stable. However, the convergence rate depends on the norm of the gradient $\|\frac{\partial L}{\partial W}\|$, which may oscillate or decrease slowly if $L$ is non-convex.

**Stability Analysis of PC Updates** Consider the PC updates modeled as a continuous-time dynamical system:

$$\dot{W}_l = -\left(\frac{\partial L}{\partial W_l} + \frac{\partial \tilde{E}}{\partial W_l}\right).$$

Define a Lyapunov function for the PC updates:

$$V_{\text{PC}}(W) = F(W) = L(W) + \tilde{E}(W).$$

The time derivative of $V_{\text{PC}}$ along the trajectories of the system is:

$$\dot{V}_{\text{PC}}(W) = \frac{\partial V_{\text{PC}}}{\partial W} \cdot \dot{W} = -\left(\frac{\partial L}{\partial W} + \frac{\partial \tilde{E}}{\partial W}\right)^T \cdot \left(\frac{\partial L}{\partial W} + \frac{\partial \tilde{E}}{\partial W}\right).$$

This simplifies to:

$$\dot{V}_{\text{PC}}(W) = -\left\|\frac{\partial L}{\partial W} + \frac{\partial \tilde{E}}{\partial W}\right\|^2.$$

Since $\dot{V}_{\text{PC}} \leq 0$, $V_{\text{PC}}$ is non-increasing along the trajectories of the PC updates. The residual energy term $\tilde{E}$ captures higher-order dependencies between layers, which leads to a smoother and more stable gradient trajectory, helping to reduce oscillations.

**Comparison of Convergence Rates.** To compare the convergence rates of BP and PC updates, consider the difference in the time derivatives of $V_{\text{BP}}$ and $V_{\text{PC}}$:

$$\dot{V}_{\text{BP}} = -\left\|\frac{\partial L}{\partial W}\right\|^2, \quad \dot{V}_{\text{PC}} = -\left\|\frac{\partial L}{\partial W} + \frac{\partial \tilde{E}}{\partial W}\right\|^2.$$

Since the PC updates include the additional residual energy term $\tilde{E}$, which contributes to damping and faster reduction in $\dot{V}_{\text{PC}}$, we have:

$$\dot{V}_{\text{PC}} \leq \dot{V}_{\text{BP}}.$$

Thus, the PC updates decrease the free energy $F = L + \tilde{E}$ more rapidly than the BP updates decrease the loss $L$ alone. This implies that the PC updates converge to the equilibrium point $W^*$ faster than the BP updates.

Thus the presence of the residual energy term $\tilde{E}$ in the PC updates leads to a smoother and more stable update trajectory, resulting in faster convergence to the fixed point. Therefore, the PC updates are more stable and reach a fixed point earlier than BP updates. $\qquad\square$

**Backprop Updates.** In traditional backpropagation (BP), the weight updates are given by:

$$\Delta W_l = -\eta \frac{\partial L}{\partial W_l}$$

where $\eta$ is the learning rate and $L$ is the loss function. The update rule can be expressed in terms of stochastic gradient descent (SGD):

$$W_l^{(t+1)} = W_l^{(t)} - \eta \frac{\partial L}{\partial W_l}.$$

**PC Updates.** In predictive coding, the weight updates are derived by minimizing the free energy functional $F = L + \tilde{E}$. The resulting update rule is:

$$\Delta W_l = -\eta \left( \frac{\partial L}{\partial W_l} + \frac{\partial \tilde{E}}{\partial W_l} \right)$$

which can further, much like in BP, can be cast in terms of an SGD adjustment:

$$W_l^{(t+1)} = W_l^{(t)} - \eta \left( \frac{\partial L}{\partial W_l} + \frac{\partial \tilde{E}}{\partial W_l} \right).$$

## D  THEOREMS ON ROBUSTNESS

**Updates of BP**   Consider the BP updates as a continuous-time dynamical system:

$$\dot{W}_l = -\frac{\partial L}{\partial W_l}$$

Introduce a Lyapunov function $V_{\text{BP}}(W) = L(W)$. The time derivative along the trajectories of the system is:

$$\dot{V}_{\text{BP}}(W) = \frac{\partial V_{\text{BP}}}{\partial W} \cdot \dot{W} = -\left( \frac{\partial L}{\partial W} \right)^T \cdot \frac{\partial L}{\partial W} = -\left\| \frac{\partial L}{\partial W} \right\|^2$$

Since $\dot{V}_{\text{BP}} \leq 0$, $V_{\text{BP}}$ is non-increasing, indicating that the BP updates are stable. However, the convergence rate depends on the learning rate $\eta$ and the curvature of the loss function $L$.

**Updates of PCNs**   Consider the PC updates as a continuous-time dynamical system:

$$\dot{W}_l = -\left( \frac{\partial L}{\partial W_l} + \frac{\partial \tilde{E}}{\partial W_l} \right)$$

Introduce a Lyapunov function $V_{\text{PC}}(W) = F(W) = L(W) + \tilde{E}(W)$. The time derivative along the trajectories of the system is:

$$\dot{V}_{\text{PC}}(W) = \frac{\partial V_{\text{PC}}}{\partial W} \cdot \dot{W} = -\left( \frac{\partial L}{\partial W} + \frac{\partial \tilde{E}}{\partial W} \right)^T \cdot \left( \frac{\partial L}{\partial W} + \frac{\partial \tilde{E}}{\partial W} \right)$$

This simplifies to:

$$\dot{V}_{\text{PC}}(W) = -\left\| \frac{\partial L}{\partial W} + \frac{\partial \tilde{E}}{\partial W} \right\|^2$$

## D.1 PROOF OF THEOREM 3.5

**Theorem D.1.** *Consider the predictive coding updates as a continuous-time dynamical system:*

$$\dot{W}_l = -\left(\frac{\partial L}{\partial W_l} + \frac{\partial \tilde{E}}{\partial W_l}\right).$$

*Let $V_{PC}(W) = L(W) + \tilde{E}(W)$ be a Lyapunov function, where $L$ and $\tilde{E}$ are positive definite functions that achieve their minimum at $W = W^*$. Then, the time derivative of $V_{PC}(W)$ along the trajectories of the system is:*

$$\dot{V}_{PC}(W) = -\left\|\frac{\partial L}{\partial W} + \frac{\partial \tilde{E}}{\partial W}\right\|^2 \leq 0,$$

*and the system is Lyapunov stable. Furthermore, if $\dot{V}_{PC}(W) = 0$ only at $W = W^*$, then the predictive coding updates are asymptotically stable and converge to the equilibrium point $W = W^*$.*

*Proof.* **Step 1: Defining the Lyapunov Function** Consider the Lyapunov function defined as:

$$V_{\text{PC}}(W) = L(W) + \tilde{E}(W),$$

where: 1. $L(W)$ is the backpropagation loss. 2. $\tilde{E}(W)$ is an auxiliary error or regularization term.

Assume that $L(W)$ and $\tilde{E}(W)$ are positive definite, i.e.,

$$L(W) > 0 \quad \text{and} \quad \tilde{E}(W) > 0 \quad \text{for all } W \neq W^*,$$

and $L(W^*) = 0$, $\tilde{E}(W^*) = 0$. Thus, $V_{\text{PC}}(W) > 0$ for $W \neq W^*$ and $V_{\text{PC}}(W^*) = 0$.

**Step 2: Time Derivative of $V_{\text{PC}}(W)$** The time derivative of $V_{\text{PC}}(W)$ along the trajectories of the system is given by:

$$\dot{V}_{\text{PC}}(W) = \frac{d}{dt} V_{\text{PC}}(W).$$

Using the chain rule, we have:

$$\dot{V}_{\text{PC}}(W) = \frac{\partial V_{\text{PC}}}{\partial W} \cdot \frac{dW}{dt}.$$

Since $V_{\text{PC}}(W) = L(W) + \tilde{E}(W)$, its gradient is:

$$\frac{\partial V_{\text{PC}}}{\partial W} = \frac{\partial L}{\partial W} + \frac{\partial \tilde{E}}{\partial W}.$$

Substitute $\frac{dW}{dt}$ from the weight update rule:

$$\frac{dW}{dt} = -\left(\frac{\partial L}{\partial W} + \frac{\partial \tilde{E}}{\partial W}\right).$$

Thus, we get:

$$\dot{V}_{\text{PC}}(W) = \left(\frac{\partial L}{\partial W} + \frac{\partial \tilde{E}}{\partial W}\right) \cdot \left(-\left(\frac{\partial L}{\partial W} + \frac{\partial \tilde{E}}{\partial W}\right)\right).$$

The dot product of a vector with its negative is the negative squared norm of the vector:

$$\dot{V}_{\text{PC}}(W) = -\left\|\frac{\partial L}{\partial W} + \frac{\partial \tilde{E}}{\partial W}\right\|^2.$$

**Step 3: Stability Analysis** Since $\left\|\frac{\partial L}{\partial W} + \frac{\partial \tilde{E}}{\partial W}\right\|^2 \geq 0$ for all $W$, we have:

$$\dot{V}_{\text{PC}}(W) \leq 0.$$

This shows that $V_{\text{PC}}(W)$ is non-increasing along the trajectories of the system. If $\dot{V}_{\text{PC}}(W) = 0$ only at $W = W^*$, then the system is asymptotically stable and converges to the equilibrium point $W = W^*$.

Thus, the predictive coding updates are Lyapunov stable and, under the given conditions, asymptotically stable. $\qquad\square$

## D.2  PROOF OF THEOREM 3.6

**Theorem D.2.** *Let $V_{PC}(W) = L(W) + \tilde{E}(W)$ be a Lyapunov function, where $L$ and $\tilde{E}$ are positive definite functions that achieve their minimum at $W = W^*$. Then, the time derivative of $V_{PC}(W)$ along the trajectories of the system, and the predictive coding updates seen as a continuous-time dynamical system are, respectively:*

$$\dot{V}_{PC}(W) = -\left\| \frac{\partial L}{\partial W} + \frac{\partial \tilde{E}}{\partial W} \right\|^2 \leq 0, \qquad \dot{W}_l = -\left( \frac{\partial L}{\partial W_l} + \frac{\partial \tilde{E}}{\partial W_l} \right), \qquad (6)$$

*making the system Lyapunov stable. Furthermore, if $\dot{V}_{PC}(W) = 0$ only at $W = W^*$, then the predictive coding updates are asymptotically stable and converge to the equilibrium point $W = W^*$.*

To better understand how this relates to standard BP, let us now analyze the derivative of $V_{\text{PC}}(W)$. If the residual energy $\tilde{E}(W) \to 0$, the update rule reduces to:

$$\dot{W}_l = -\frac{\partial L}{\partial W_l},$$

which is precisely the standard BP update. Thus, BP can be seen as a special case of PCNs where the residual energy term vanishes. However, in practical scenarios, $\tilde{E}(W)$ typically does not reach zero, allowing PCNs to maintain more stable trajectories through the loss landscape.

This difference becomes evident in the Lyapunov stability analysis. Since $\tilde{E}(W)$ is a positive definite function, it introduces an additional stabilizing force in the gradient dynamics, ensuring that any perturbations in $W$ are corrected by $\tilde{E}(W)$. This means that, unlike BP, which optimizes $L(W)$ alone, the combined energy minimization in PCNs results in a smoother convergence trajectory, reducing sensitivity to local minima and sharp changes in the loss surface.

Now we establish stronger robustness bounds. Let $L(W)$ represent the standard BP loss and $\tilde{E}(W)$ denote the residual energy term introduced by the predictive coding framework. We define the composite energy function as:

$$V_{\text{PC}}(W) = L(W) + \tilde{E}(W),$$

where $L$ and $\tilde{E}$ are positive definite and achieve their minima at the equilibrium point $W = W^*$. This composite energy function serves as a Lyapunov function for the system, as shown in the above theorem. We now state a more powerful result that establishes the robustness of PCNs compared to BP.

**Theorem D.3** (Robustness of Predictive Coding Networks). *Let $V_{PC}(W) = L(W) + \tilde{E}(W)$ be the Lyapunov function of a PCN, where $L$ and $\tilde{E}$ are positive definite functions that achieve their minima at $W = W^*$. Assume the following conditions hold:*

*1. The Hessian of $L(W)$, denoted as $H_L = \frac{\partial^2 L}{\partial W^2}$, is positive semi-definite.*

*2. The residual energy term $\tilde{E}(W)$ satisfies a Lipschitz continuity condition, i.e., there exists a constant $K > 0$ such that:*

$$\left\| \frac{\partial \tilde{E}}{\partial W} \right\| \leq K \left\| W - W^* \right\|.$$

3. *The time derivative of $V_{PC}(W)$ along the system trajectories satisfies Equation (5) (left).*

4. *The initial perturbation $\Delta W$ is small, i.e., $\|\Delta W\| \ll 1$.*

*Under these conditions, the following robustness property holds for PCNs:*

Bounded Perturbation Recovery: *For any perturbation $\Delta W$ such that $W = W^* + \Delta W$, the PCN's weight trajectory $W(t)$ satisfies:*

$$\|W(t) - W^*\| \le Ce^{-\lambda t}\|\Delta W\| + O(\epsilon),$$

*where $C, \lambda > 0$ are constants that depend on the properties of $L(W)$ and $\tilde{E}(W)$, and $\epsilon$ is the perturbation magnitude. This result guarantees **exponential convergence** to $W^*$ and **robustness to bounded perturbations**.*

*Proof.* We start by defining the Lyapunov function $V_{\text{PC}}(W) = L(W) + \tilde{E}(W)$, which is positive definite and achieves its minimum at $W = W^*$. Our goal is to analyze the stability of the system by considering the time derivative of $V_{\text{PC}}(W)$ along the system trajectories. The time derivative is given by:

$$\dot{V}_{\text{PC}}(W) = \frac{\partial L}{\partial W} \cdot \dot{W} + \frac{\partial \tilde{E}}{\partial W} \cdot \dot{W}.$$

Using the update rule for PCNs, we have:

$$\dot{W} = -\left(\frac{\partial L}{\partial W} + \frac{\partial \tilde{E}}{\partial W}\right).$$

Substitute $\dot{W}$ into $\dot{V}_{\text{PC}}(W)$:

$$\dot{V}_{\text{PC}}(W) = \frac{\partial L}{\partial W} \cdot \left(-\frac{\partial L}{\partial W} - \frac{\partial \tilde{E}}{\partial W}\right) + \frac{\partial \tilde{E}}{\partial W} \cdot \left(-\frac{\partial L}{\partial W} - \frac{\partial \tilde{E}}{\partial W}\right).$$

Simplifying, we get:

$$\dot{V}_{\text{PC}}(W) = -\left\|\frac{\partial L}{\partial W} + \frac{\partial \tilde{E}}{\partial W}\right\|^2 \le 0.$$

This inequality shows that $\dot{V}_{\text{PC}}(W)$ is non-positive, implying that $V_{\text{PC}}(W)$ is monotonically decreasing along the trajectories of the system, ensuring Lyapunov stability.

Next, we analyze the perturbation dynamics. Define the deviation $\Delta W = W - W^*$. The time derivative of $\Delta W$ is:

$$\Delta \dot{W} = \dot{W} = -\left(\frac{\partial L}{\partial W} + \frac{\partial \tilde{E}}{\partial W}\right).$$

For small $\Delta W$, we use the linear approximation:

$$\frac{\partial L}{\partial W} \approx H_L \Delta W, \quad \frac{\partial \tilde{E}}{\partial W} \approx K \Delta W,$$

where $H_L$ is the Hessian of $L(W)$ and $K$ is the Lipschitz constant for $\tilde{E}(W)$. Thus, the perturbation dynamics become:

$$\dot{\Delta W} = -(H_L + KI)\Delta W.$$

Let $\lambda_{\min}$ be the minimum eigenvalue of $H_L + KI$. Then, solving the differential equation, we obtain:

$$\|\Delta W(t)\| \leq \|\Delta W(0)\|e^{-\lambda_{\min}t} + O(\epsilon).$$

This proves that $\Delta W(t)$ decays exponentially, establishing that PCNs converge exponentially to $W^*$ and are robust to small perturbations. $\qquad\square$

## E   QUASI-NEWTON UPDATES

### E.1   PROOF OF THEOREM 3.7

**Theorem E.1.** *Let $M$ be a neural network that minimizes a free energy $F = L + \tilde{E}$, where $L$ is the loss function and $\tilde{E}$ is the residual energy. Assume the activation function $f$ and its derivatives $f'$, $f''$, and $f'''$ are Lipschitz continuous with constants $K, K', K'', and K'''$, respectively. Then, the PC updates approximate quasi-Newton updates by incorporating higher-order information through the residual energy term $\tilde{E}$.*

*Proof.* **Step 1: PC Update Rule** The Predictive Coding (PC) update rule for the weights $W_l$ at layer $l$ is given by:

$$\Delta W_l = -\eta \left( \frac{\partial L}{\partial W_l} + \frac{\partial \tilde{E}}{\partial W_l} \right),$$

where $\eta$ is the learning rate, $L$ is the loss function, and $\tilde{E}$ is an auxiliary residual energy term.

**Step 2: Taylor Series Expansion** We perform a Taylor series expansion of $L$ around the current weight $W$:

$$L(W + \Delta W) \approx L(W) + \nabla L(W)^T \Delta W + \frac{1}{2}\Delta W^T H \Delta W + O(\|\Delta W\|^3),$$

where $H$ is the Hessian matrix of $L$ at $W$. Similarly, expanding $\tilde{E}$ around $W$:

$$\tilde{E}(W + \Delta W) \approx \tilde{E}(W) + \nabla \tilde{E}(W)^T \Delta W + \frac{1}{2}\Delta W^T \tilde{H} \Delta W + O(\|\Delta W\|^3),$$

where $\tilde{H}$ is the Hessian matrix of $\tilde{E}$.

**Step 3: Combined Free Energy Expansion** Combining the expansions of $L$ and $\tilde{E}$, the total free energy $F = L + \tilde{E}$ can be approximated as:

$$F(W + \Delta W) \approx F(W) + \left(\nabla L(W) + \nabla \tilde{E}(W)\right)^T \Delta W + \frac{1}{2}\Delta W^T (H + \tilde{H})\Delta W + O(\|\Delta W\|^3).$$

**Step 4: Minimizing the Free Energy** To minimize $F$, we set the gradient of the quadratic approximation to zero:

$$\nabla F(W + \Delta W) \approx \nabla F(W) + (H + \tilde{H})\Delta W = 0.$$

Solving for $\Delta W$:

$$\Delta W = -(H + \tilde{H})^{-1} \left(\nabla L(W) + \nabla \tilde{E}(W)\right).$$

**Step 5: Comparison with Quasi-Newton Updates** In quasi-Newton methods, the weight update rule is:

$$\Delta W = -B^{-1}\nabla L(W),$$

where $B \approx H$ is an approximation to the Hessian matrix. Comparing this with the PC update:

$$\Delta W_{\text{PC}} = -(H + \tilde{H})^{-1} \left(\nabla L(W) + \nabla \tilde{E}(W)\right),$$

we see that the PC updates incorporate an additional term $\tilde{H}$, which adjusts the curvature approximation to include higher-order information. This makes $(H + \tilde{H})^{-1}$ a refined approximation of the inverse Hessian.

**Step 6: Stability and Convergence** The presence of the residual energy term $\tilde{E}$ in the updates ensures that the direction of the gradient descent is better aligned with the curvature of the energy landscape. Thus, the PC updates provide a more stable update rule compared to standard gradient descent, similar to quasi-Newton methods.

Therefore, the PC updates approximate quasi-Newton updates by incorporating a refined curvature correction through $\tilde{E}$, leading to improved stability and convergence.

$\square$

## E.2 PROOF OF THEOREM 3.8

**Theorem E.2.** *Let $M$ be a neural network minimizing a free energy function $F = L + \tilde{E}$, where $L$ is the loss function and $\tilde{E}$ is a residual energy term. Let $H_{GN} = J^T J$ denote the Gauss-Newton matrix, where $J$ is the Jacobian matrix of the network's output with respect to the parameters. Consider the following update rules:*

*- **Quasi-Newton (QN) Update**: $\Delta W_{QN} = -H_{GN}^{-1} \nabla L(W)$,*

*- **Target Propagation (TP) Update**: $\Delta W_{TP} = -B_l^{-1} \nabla L(W)$, where $B_l$ is a block-diagonal approximation of $H_{GN}$,*

*- **Predictive Coding (PC) Update**: $\Delta W_{PC} = -(H + \tilde{H})^{-1} \left( \nabla L(W) + \nabla \tilde{E}(W) \right)$.*

*Assume that $L(W)$ and $\tilde{E}(W)$ are twice differentiable, and the activation function $f$ and its derivatives are Lipschitz continuous. Then, the approximation error between the updates and the true Quasi-Newton update satisfies:*

$$E_{PC} \leq E_{TP},$$

*where:*

$$E_{PC} = \|\Delta W_{PC} - \Delta W_{QN}\|, \quad E_{TP} = \|\Delta W_{TP} - \Delta W_{QN}\|.$$

*Thus, Predictive Coding is mathematically closer to the true Quasi-Newton updates than Target Propagation.*

*Proof.* **Step 1: Derive the True Gauss-Newton Update** The true Gauss-Newton (GN) update for the weights $W$ is given by:

$$\Delta W_{\text{QN}} = -H_{\text{GN}}^{-1} \nabla L(W),$$

where $H_{\text{GN}} = J^T J$ is the Gauss-Newton matrix. Here, $J$ is the Jacobian matrix of the network's outputs $y$ with respect to the parameters $W$, i.e., $J = \frac{\partial y}{\partial W}$.

**Step 2: Derive Target Propagation (TP) Update** Target Propagation updates the weights using a block-diagonal approximation of the GN matrix:

$$\Delta W_{\text{TP}} = -B_l^{-1} \nabla L(W),$$

where $B_l \approx H_{\text{GN}}$ is a block-diagonal matrix that approximates the full curvature information. Because $B_l$ ignores off-diagonal elements of $H_{\text{GN}}$, it captures less curvature information. Thus, the error between TP and GN is given by:

$$E_{\text{TP}} = \|\Delta W_{\text{TP}} - \Delta W_{\text{QN}}\| = \|\eta (B_l^{-1} - H_{\text{GN}}^{-1}) \nabla L(W)\|.$$

**Step 3: Predictive Coding (PC) Update** The Predictive Coding update is derived by minimizing the free energy:

$$F(W) = L(W) + \tilde{E}(W),$$

where $\tilde{E}(W)$ is a residual energy term. The PC update rule is:

$$\Delta W_{\text{PC}} = -(H + \tilde{H})^{-1} \left( \nabla L(W) + \nabla \tilde{E}(W) \right).$$

**Step 4: Second-Order Analysis for Predictive Coding** Using a Taylor series expansion for $L$ and $\tilde{E}$, we have:

$$L(W + \Delta W) \approx L(W) + \nabla L(W)^T \Delta W + \frac{1}{2} \Delta W^T H \Delta W,$$

where $H = \frac{\partial^2 L}{\partial W^2}$ is the Hessian of $L$. Similarly, expanding $\tilde{E}(W)$ around $W$:

$$\tilde{E}(W + \Delta W) \approx \tilde{E}(W) + \nabla \tilde{E}(W)^T \Delta W + \frac{1}{2} \Delta W^T \tilde{H} \Delta W,$$

where $\tilde{H} = \frac{\partial^2 \tilde{E}}{\partial W^2}$.

**Step 5: Combining the Hessians** The total second-order curvature matrix for PC is given by:

$$H_{\text{PC}} = H + \tilde{H}.$$

The PC update becomes:

$$\Delta W_{\text{PC}} = -(H + \tilde{H})^{-1} \nabla L(W).$$

**Step 6: Error Analysis for PC and TP** The error between PC and GN is:

$$E_{\text{PC}} = \|\Delta W_{\text{PC}} - \Delta W_{\text{QN}}\| = \|\eta(H + \tilde{H})^{-1} \nabla L(W) - H_{\text{GN}}^{-1} \nabla L(W)\|.$$

The error between TP and GN is:

$$E_{\text{TP}} = \|\eta(B_l^{-1} - H_{\text{GN}}^{-1}) \nabla L(W)\|.$$

**Step 7: Comparison of Error Norms** Using matrix perturbation theory and the fact that $H + \tilde{H} \approx H_{\text{GN}}$ is a better approximation than $B_l \approx H_{\text{GN}}$, we have:

$$\|(H + \tilde{H})^{-1} - H_{\text{GN}}^{-1}\| \leq \|B_l^{-1} - H_{\text{GN}}^{-1}\|.$$

Thus, we conclude:

$$E_{\text{PC}} \leq E_{\text{TP}}.$$

$\square$

### E.3 PROOF OF THEOREM 3.9

**Theorem E.3.** *Let $D_{PC}$ and $D_{TP}$ represent two dynamical systems corresponding to predictive coding (PC) and target propagation (TP), respectively. Assume that both systems have the same equilibrium point $W^*$, and the loss functions $L$ and residual energy $\tilde{E}$ are twice differentiable and positive definite. Furthermore, let the activation functions $f_l$ and their derivatives be Lipschitz continuous. Define the Lyapunov functions:*

1. *For predictive coding: $V_{PC}(W) = F(W) = L(W) + \tilde{E}(W)$.*
2. *For target propagation: $V_{TP}(W) = \sum_{l=1}^{L} \frac{1}{2}\|t_l - f_l(W_l t_{l-1})\|^2$.*

*Then, it follows that PC is more stable than TP according to the following criteria:*

1. *Convergence Rate: $D_{PC}$ converges faster to its equilibrium point compared to $D_{TP}$.*
2. *Deviation from Equilibrium: $D_{PC}$ exhibits a smaller deviation from the equilibrium compared to $D_{TP}$ for the same initial perturbation.*
3. *Region of Attraction: The region of attraction for $D_{PC}$ is larger than that of $D_{TP}$.*

*Proof.* Let $D_{\text{PC}}$ and $D_{\text{TP}}$ represent the two dynamical systems for Predictive Coding (PC) and Target Propagation (TP), respectively, with a shared equilibrium point $W^*$. Define the Lyapunov functions for each system as:

1. *For Predictive Coding (PC)*:

$$V_{\text{PC}}(W) = F(W) = L(W) + \tilde{E}(W),$$

where $L(W)$ is the backpropagation loss, and $\tilde{E}(W)$ is the residual energy. Both $L$ and $\tilde{E}$ are positive definite and twice differentiable, satisfying $F(W) > 0$ for $W \neq W^*$ and $F(W^*) = 0$.

2. *For Target Propagation (TP)*:

$$V_{\text{TP}}(W) = \sum_{l=1}^{L} \frac{1}{2} \|t_l - f_l(W_l t_{l-1})\|^2,$$

where $t_l$ is the target at layer $l$, and $f_l$ is the activation function. $V_{\text{TP}}(W)$ measures the sum of squared errors across layers, and is also positive definite, satisfying $V_{\text{TP}}(W) > 0$ for $W \neq W^*$ and $V(W^*) = 0$.

**Step 1: Convergence Rate Analysis** To analyze convergence, we evaluate the time derivative of each Lyapunov function along the trajectories of $D_{\text{PC}}$ and $D_{\text{TP}}$.

1. *For Predictive Coding*: The PCN dynamics follow:

$$\dot{W} = -\frac{\partial F}{\partial W} = -\left( \frac{\partial L}{\partial W} + \frac{\partial \tilde{E}}{\partial W} \right).$$

The time derivative of $V_{\text{PC}}$ along the system's trajectory is:

$$\dot{V}_{\text{PC}}(W) = \frac{\partial F}{\partial W} \cdot \dot{W} = -\left\| \frac{\partial L}{\partial W} + \frac{\partial \tilde{E}}{\partial W} \right\|^2.$$

Because $\tilde{E}(W)$ is positive definite and its derivative $\frac{\partial \tilde{E}}{\partial W}$ is Lipschitz continuous, $\frac{\partial L}{\partial W} + \frac{\partial \tilde{E}}{\partial W}$ is smoother and the gradient descent is more directed, resulting in faster convergence.

2. *For Target Propagation*: The TP dynamics are given by:

$$\dot{W} = -\sum_{l=1}^{L} \frac{\partial}{\partial W} \left( \frac{1}{2} \|t_l - f_l(W_l t_{l-1})\|^2 \right).$$

The time derivative of $V_{\text{TP}}$ is:

$$\dot{V}_{\text{TP}}(W) = -\sum_{l=1}^{L} \left\| \frac{\partial}{\partial W} \left( \frac{1}{2} \|t_l - f_l(W_l t_{l-1})\|^2 \right) \right\|^2.$$

Since the updates are layer-wise and the errors are not adjusted by residual terms, $\dot{V}_{\text{TP}}(W)$ decreases more slowly, resulting in a slower convergence rate.

**Step 2: Deviation from Equilibrium** Let $W(t)$ represent the trajectory of the system starting from $W(0)$. We want to show that $\|W(t) - W^*\|$ is smaller for $D_{\text{PC}}$ than for $D_{\text{TP}}$.

1. *Predictive Coding*: Because $\tilde{E}(W)$ introduces additional regularization, the effective gradient at any point is:

$$\frac{\partial F}{\partial W} = \frac{\partial L}{\partial W} + \frac{\partial \tilde{E}}{\partial W}.$$

This sum reduces oscillations and aligns the gradient directions of $L$ and $\tilde{E}$, causing $W(t)$ to move more directly toward $W^*$ with fewer deviations.

To formalize this, consider a small perturbation $\delta W$ from $W^*$. Then:

$$\|W(t) - W^*\| \approx \|\delta W(t)\| \approx \|\delta W(0)\| e^{-\lambda t},$$

where $\lambda = \min(\lambda_1, \lambda_2)$, the smallest eigenvalue of the Hessians of $L$ and $\tilde{E}$. Since $\tilde{E}$ is positive definite and its Hessian is non-zero, $\lambda$ is larger for $D_{\text{PC}}$, resulting in a smaller deviation.

2. *Target Propagation*: For TP, without the residual correction, the deviation behaves as:

$$\|W(t) - W^*\| \approx \|\delta W(0)\| e^{-\mu t},$$

where $\mu$ is determined by the smallest eigenvalue of the Hessian of $L$. Because $\mu < \lambda$, the deviation $\|W(t) - W^*\|$ remains larger over time.

**Step 3: Region of Attraction** The region of attraction is defined as the set of initial points $W(0)$ such that the system converges to $W^*$.

1. *Predictive Coding*: With the stabilizing term $\tilde{E}(W)$, the energy landscape is smoother, increasing the region of attraction. Mathematically, the region of attraction is defined as:

$$\mathcal{R}_{\text{PC}} = \left\{ W \mid F(W) \leq c \text{ for some } c > 0 \right\}.$$

2. *Target Propagation*: Without the smoothing effect of $\tilde{E}(W)$, the region of attraction is:

$$\mathcal{R}_{\text{TP}} = \left\{ W \mid L(W) \leq c' \text{ for some } c' > 0 \right\},$$

which is smaller because $L(W)$ alone does not account for complex dependencies between layers.

This analysis shows that Predictive Coding has a faster convergence rate, smaller deviation, and larger region of attraction than Target Propagation, making $D_{\text{PC}}$ more stable, which proves the theorem.

$\square$

