# OpenReview forum: "Tight Stability, Convergence, and Robustness Bounds for Predictive Coding Networks"
_ICLR.cc/2025/Conference — Submitted to ICLR 2025_

### Official Review · Reviewer_wf38 · 2024-10-31

**Soundness:** 2
**Presentation:** 2
**Contribution:** 1
**Rating:** 3
**Confidence:** 3

**Summary:**

In this paper, the authors study the stability, convergence, and robustness of predictive coding networks (PCNs) from a dynamical systems perspective. They demonstrate that:

* PCNs are stable in both the inference and training phases and exhibit greater stability than backpropagation (BP);
* PCNs are robust to perturbations in $W$ in the training phase;
* The Quasi-Newton update in PCNs is more stable than the traditional training procedure (TP).

Numerical experiments further highlight the differences between PC and BP.

**Strengths:**

The idea of using dynamic systems theory in PC is interesting. The authors reformulate the theorems in natural language instead of math to make the technical paper more accessible.

**Weaknesses:**

I find the analysis somewhat constrained, with the mathematical theorems not strong enough to fully support the authors’ claims. The content of the paper, in my view, does not quite meet the expectations set by the title and abstract. To be more specific:

* I have the impression that all the analyses are based on the continuous-time approximation of the discrete update rule. This should be justified, and may have an impact on the proofs. For example, with discrete update rule, where overshoot is possible, does V(x) still decrease along the path?
* It appears to me the authors assume implicitly that there is only one minimum $x*$/$W*$, which may not be the case in reality.
* In the appendix, the authors characterize robustness as Lipschitz continuity (example 2 on pages 17-18) with respect to input data. However, the Lipschitz continuity of PCN with respect to $x$ is untouched in the robustness discussion.
* I find the robustness and stability of PCN particularly interesting, as the loss function is optimized with respect two sets of parameters. If I understand correctly, the output of all intermediate layers $x_{1, …, L-1}$ are calculated as the argmin of $F$ featuring the initial weights. Then, the weights are updated by minimizing the $F$ featuring $x_{1, …, L-1}$. In principle, $x_{1, …, L-1}$ contains the information of initial weights, which could be passed into the final $W$. To make it more interesting, argmin is not a continuous function, and perturbation in initial $W$ may have a big impact on $x_{1, …, L-1}$ and final $W$. However, none of the theorems in the paper is dedicated to the interplay between minimizations with respect to $W$ and $x$.
* I feel that I miss the point in the section ​​PREDICTIVE CODING UPDATES AS QUASI-NEWTON UPDATES. It appears to me that the update rule for PC in this section is not the same as eq (4). So the authors are comparing the QN version of PC update to QN, instead of equation (4) to QN. I am wondering how the discussion here could support "PC is significantly closer to quasi-Newton updates than TP".

Minor:
* The beginning of section 3 is not well organized. Maybe it is better to group theorem 3.1-3.4 into a subsection.
* Fig 3, the text in the figure overlaps.

**Questions:**

In theorem 3.1, the authors expand the function $f$ to the third power of $W-W*$. Is there a specific reason for choosing order 3?

I wonder if the authors have comment regarding the weaknesses.

---

### Official Review · Reviewer_fVqr · 2024-11-01

**Soundness:** 1
**Presentation:** 3
**Contribution:** 2
**Rating:** 3
**Confidence:** 3

**Summary:**

Predictive coding networks (PCNs) are layered neural networks trained by minimizing a free energy function consisting in a sum of contributions from each layer of neurons (local energies), putting a positive cost on the distance between the neural activity of the layer and the post-activation of the previous layer. The paper under review considers a training procedure in subsequent steps: first the hidden neural activities are updated to convergence by minimizing this free energy at fixed input, output and weights (inference phase, E-steps); then the weights are updated, minimizing the same functional evaluated at the previously obtained value of the hidden neuronal activities (learning phase, M-steps).

By leveraging a mapping from PCNs to continuous time dynamical systems, under assumptions on the activation function of the hidden layers (Lipschitz continuity of itself and its first 3 derivatives), the paper under review proves theorems on the convergence and stability of the inference phase (Theorems 3.1, 3.2 and 3.3); convergence of the learning phase at a faster rate (in terms of parameter updates) than standard backpropagation (Theorem 3.4); robustness (exponential convergence to a fixed point) of the learning phase (Theorem 3.5 and 3.6); approximate quasi-Newton nature of the learning steps, and comparison with target propagation (Theorems 3.7, 3.8 and 3.9). The paper also provides numerical evidence of comparable performances and faster convergence rates of PCN against backpropagation and target propagation when training a custom convolutional network on MNIST, fashion-MNIST, CIFAR10, CIFAR100.

**Strengths:**

The idea of predictive coding networks, thought not novel, is intriguing and sufficiently well explained in the paper (Section 2). Lyapunov's theory of dynamical systems is also well introduced (Section 2.1). The idea of applying concepts from dynamical systems to prove convergence, stability and robustness properties of PCNs is sound and, as far as I know, novel in this context, thought I am not an expert on the literature for this specific topic. I think that a proof of convergence of the PCN protocol, and a comparison with standard BP or other methods, would be extremely interesting for the community. I am fairly confident that the theorems on the inference phase alone (3.1, 3.2 and 3.3) are relevant. I think the same of the theorems proving that the weight updates of PCN are approximating quasi-Newton updates (3.7, 3.8). The numerical experiments support some of the theoretical claims of the paper and are performed with relevant models and datasets.

**Weaknesses:**

The paper adopts an algorithmic-agnostic perspective when it comes to the way the gradients are computed during the learning phase. This aspect is crucial to prove the theorems of convergence of the learning phase, which are based on the assumption (never clearly stated) that the discretized dynamics of the weights is well described by a continuous-time deterministic dynamical system (see for example Appendix C.4). It is known that SGD updates in the case of normal BP are approximated by a stochastic dynamical system, meaning that the discrete-time trajectories remain close in expectation to the ones of a continuous stochastic differential equation, accounting for the noise that the random choice of input points in the batches introduces in the system, and for times that do not exceed the inverse of the learning rate. Different protocols (constant rate, Nesterov's, momentum, etc) usually require different proofs of convergence.

The paper proves, at best, convergence results on the continuous-time deterministic counterpart of PCNs. Notice that, while for generic discrete maps the mapping to a deterministic gradient flow can be justified for small learning rate under the assumption that the gradients are known exactly, the SGD nature is intrinsic in PCNs, where, for each datapoint, first the hidden neural activities are updated to convergence and then the weights are updated. This and other minor weaknesses are detailed below directly in the Questions section. I think that this concern alone is relevant enough to support an overall grade of reject to the paper, as I feel that the authors need a substatial amount of work to overcome it, or to reduce their claims and the significance of the paper.

**Questions:**

- The free energy function reported in Eq. 1 is a function of the hidden neural activities $\\{x_l\\}$, the input data $x_0$, the target data $x_L$ and the weights of the network. Theorems 3.1-3.3 prove convergence of the inference phase to a fixed point $x^*=\\{x_l^*\\}$. The inference phase happens at fixed weights, input and target data, which means that, in general, $x^* = x^*(W,x_0,x_L)$. Then, the weights are updated, new input and target point are chosen from the training set, and the whole process is repeated up to convergence of the weight updates (this whole process is not completely explicit from the paper, see below, but can be found in literature, see for example Salvatori et al, 2023). This means that, at each step of the weights update, the fixed point $x^*$ to which the inference phase is converging is different, and the update of the weights is noisy depending on how the input and output data are selected from the training set. This is a well known property of SGD, that can be approximated by a stochastic differential equation in continuous time with noise induced by the random choice of data from the training set. The details of this continuous stochastic counterpart depend on the particular scheme of SGD adopted (time dependent rates, batch sizes, etc). See for example [Laborde, Oberman 2020]. I do not see how the authors can derive results on convergence of the whole training phase by mapping the discrete weight updates to a *deterministic* gradient flow, see the proof of theorem 3.4 etc. Can the authors prove that the training of PCNs, as described in Eq. (3), (4), converges to a fix point under the regularity hypotheses they chose, as claimed in the paper, rather than only prove that its continuous-time deterministic counterpart does?

Ref: Laborde, Oberman. A Lyapunov analysis for accelerated gradient methods: from deterministic to stochastic case, AISTATS 2020

- If the answer to the previous question is yes, under which hypotheses on learning rate, minibatch size, etc?

- The only scenario I can currently think of, for which the authors' derivation is possibly motivate, is if the fixed point $x^*$ does not depend on $x_0$, $x_L$, and so on the choice of the specific input and target point in the training set. I have the impression that this assumption trivializes the dynamics, as the relaxation phase leaves no memory of the training set in the process. Do the authors agree on this point?

- If I understood correctly the PCN paradigm, equation (3) (to convergence) and equation (4) (single or few steps) are repeatedly alternated picking each time a random training point, and sweeping the full training set for multiple epochs. Is this the case, as described in previous literature? Can the authors clarify this aspect more clearly around equation (3) and (4)?

- I found initially hard to understand which theorems deal with convergence of the inference phase, and which ones deal with convergence of the full training phase. This is completely clear only from the proofs in Appendix. Can the author specify better the scope of each theorem? This can be done, for example, stating the full dependence of the free energy function as $F(x,x_0,x_L, W)$, and clarifying which variations are considered each time, and which arguments are kept fixed.

- From a bird-eye reading of the proofs (C.2-C.3), I do not understand the difference between theorem 3.2 and 3.3. Can the authors clarify?

- Some typos:
- Sec.~2.1: ``notably used to be formally prove the theorems of this work''
- Table 1: the bold highlighting the best test error in the table is missing.

---

### Official Review · Reviewer_c72i · 2024-11-03

**Soundness:** 3
**Presentation:** 2
**Contribution:** 2
**Rating:** 5
**Confidence:** 3

**Summary:**

The paper explores the properties of PCNs by leveraging the framework of dynamical systems and Lyapunov stability theory. It demonstrates that, under certain conditions, the energy function in PCNs can be used as a Lyapunov function, allowing the dynamics of PCNs to be expressed as a continuous-time dynamical system. The findings include: (i) the PCN updates are Lyapunov stable and converge to the equilibrium point faster than BP updates; (ii) the PCN dynamics are robust to initial bounded perturbations; (iii) the PC updates approximate quasi-Newton updates by incorporating higher-order information through the residual energy term, and is closer to the true Quasi-Newton updates compared to TP; (iv) PC is more stable than TP.

**Strengths:**

The paper analyzes the stability, robustness, and convergence of PC, and provides a theoretical comparison with BP and TP. This establishes a solid theoretical foundation for PC and offers insights into its advantages relative to other learning schemes.

**Weaknesses:**

The paper presents several theorems (Theorems 3.1, 3.2, 3.3, 3.4, 3.5, and 3.6) that appear to convey similar results, leading to redundancy in the statements. Is there a more effective way to combine and present these findings?

**Questions:**

How does the runtime of PC compared to that of BP? Under what circumstances is PC preferred over BP, or is it always the better choice? What do the plots labeled C and D in each figure indicate?

---

### Official Review · Reviewer_XBzE · 2024-11-04

**Soundness:** 2
**Presentation:** 2
**Contribution:** 2
**Rating:** 5
**Confidence:** 2

**Summary:**

This paper analyzes stability, robustness, and convergence of predictive coding (PC). The authors establish stability, convergence and robustness of the algorithm to bounded random disturbances. Moreover, the authors show that PC approximates quasi-Newton updates which leads to faster convergence when compared to Back Propagation (BP) and Target Propagation (TP). A few numerical results are included to illustrate some of the theory developed.

**Strengths:**

The results of the paper are interesting and encouraging. However, the reviewer is not familiar with the past literature on PCNs so it was hard to evaluate the novelty of the results (see weaknesses).

The writing is clear and no grammatical errors were found. The reviewer did not have time to carefully check the soundness of the technical proofs.

**Weaknesses:**

- Definition 2.1 refers to Lyapunov stability as stability. However, Definition 2.7 refers to an analogue of Definition 2.7 as Lyapunov stability. I would encourage the authors to stick to standard dynamical system terminology and refer to Definition 2.1 as Lyapunov stability. Also, I would  encourage the authors to provide references to each of their definitions.

- The exposition of the results of the paper is not great. In particular, the reviewer believes that the number of theorems included in the  paper makes it hard to follow: the theorems are repetitive so I encourage the authors provide a list of assumptions to avoid that. Moreover, the conclusions are vague. One example is the phrase "the convergence and stability of the PCN can be characterized by the bounds involving these higher-order derivatives" in Thm. 3.1. Why not include an equation to exemplify this?

- The paper lacks a proper literature review and the results are not well contextualized. The reviewer is not familiar with the past literature on PCNs so it was hard to evaluate the novelty of the results.

-Labels and axes on Figure 3 are hard to read.

**Questions:**

- What is the meaning of "a PCN that minimizes a free energy"? Why is that included? So that there exists an equilibrium for the dynamical system?

- The accuracy of PC seems to be worst than TP based upon the results shown in Figure 2. Doesn't that contradict Thm. 3.9? Can the authors comment on that?

- How realistic is the assumption that the perturbations of the algorithm are bounded like the assumptions in Thm. 3.6?

---

### Meta-Review · Area_Chair_8r4E · 2024-12-19

**Metareview:**

**Summary of Discussion:**
The reviewers found the theoretical contributions of the paper interesting but insufficiently substantiated, with significant concerns about clarity, rigor, and practicality. Key criticisms include:

1. **Over-reliance on Continuous-Time Approximation:** The analysis heavily depends on continuous-time dynamics, which may not align with the stochastic nature of real-world updates (e.g., SGD). This gap raises doubts about the applicability of the stability and convergence results to actual training scenarios.

2. **Insufficient Justification for Assumptions:** Several assumptions, including strong convexity of the free energy function and gradient flow approximation, were inadequately motivated or not grounded in practical considerations.

3. **Inadequate Novelty and Practical Relevance:** The reviewers noted a lack of substantial novelty, with many results being incremental or relying on existing techniques. The practical utility of predictive coding networks (PCNs) relative to other methods like backpropagation (BP) remains unclear.

4. **Presentation Issues:** The paper was noted for being difficult to follow, with redundant theorems, inconsistent terminology, and vague descriptions. Key insights, such as the claim that PCN updates resemble quasi-Newton methods, were not convincingly demonstrated.

5. **Rebuttal Limitations:** While the authors addressed some concerns during the rebuttal, the extensive changes required for the theoretical framework and presentation suggest the paper would benefit from a resubmission.

**Conclusion:**
The paper has potential but falls short of the standards for acceptance at ICLR. A major revision addressing the concerns about clarity, rigor, and practical relevance is necessary before the paper can be reconsidered.

**Additional Comments On Reviewer Discussion:**

See above

---

### Decision · Program_Chairs · 2025-01-22

Reject